# Regulation of pulmonary surfactant by the adhesion GPCR GPR116/ADGRF5 requires a tethered agonist-mediated activation mechanism

**James P Bridges[1†], Caterina Safina[2], Bernard Pirard[2], Kari Brown[1†], Alyssa Filuta[1†], Ravichandran Panchanathan[3], Rochdi Bouhelal[2], Nicole Reymann[2], Sejal Patel[4], Klaus Seuwen[2], William E Miller[3], Marie-Gabrielle Ludwig[2]***

[1]Department of Pediatrics, Perinatal Institute, Section of Pulmonary Biology, Cincinnati Children's Hospital Medical Center, Cincinnati, United States; [2]Novartis Institutes for Biomedical Research, Basel, Switzerland; [3]Department of Molecular Genetics, Biochemistry and Microbiology, University of Cincinnati College of Medicine, Cincinnati, United States; [4]Novartis Institutes for Biomedical Research, Cambridge, United States

**\*For correspondence:**
marie-gabrielle.ludwig@novartis.com

**Present address:** [†]Department of Medicine, Division of Pulmonary, Critical Care and Sleep Medicine, National Jewish Health, Denver, United States

**Abstract** The mechanistic details of the tethered agonist mode of activation for the adhesion GPCR ADGRF5/GPR116 have not been completely deciphered. We set out to investigate the physiological importance of autocatalytic cleavage upstream of the agonistic peptide sequence, an event necessary for NTF displacement and subsequent receptor activation. To examine this hypothesis, we characterized tethered agonist-mediated activation of GPR116 in vitro and in vivo. A knock-in mouse expressing a non-cleavable GPR116 mutant phenocopies the pulmonary phenotype of GPR116 knock-out mice, demonstrating that tethered agonist-mediated receptor activation is indispensable for function in vivo. Using site-directed mutagenesis and species-swapping approaches, we identified key conserved amino acids for GPR116 activation in the tethered agonist sequence and in extracellular loops 2/3 (ECL2/3). We further highlight residues in transmembrane 7 (TM7) that mediate stronger signaling in mouse versus human GPR116 and recapitulate these findings in a model supporting tethered agonist:ECL2 interactions for GPR116 activation.

## Editor's evaluation

Adhesion GPCRs are a relatively understudied GPCR family. One of the mechanisms they are activated by is a tethered agonist peptide that interacts with the transmembrane domain to activate the receptor. This work first shows that a knock-in mouse expressing a non-cleavable GPR116 mutant to prevent the release of the tethered agonist peptide phenocopies the pulmonary phenotype of GPR116 knock-out mice, demonstrating that tethered agonist-mediated receptor activation is indispensable for GPR116 function in vivo. The study then uses mutagenesis and activity assays to find residues in the tethered agonist that are most important for receptor activation, as well as mutating loops in the extracellular face of the receptor to find residues important in mediating the response to the tethered agonist.

## Introduction

Adhesion G protein-coupled receptors (aGPCRs) are critical for a variety of biological and pathophysiological processes, including modulation of metabolic (*Olaniru and Persaud, 2019*) and immune responses (*Lin et al., 2017*) to regulatory roles in the central and peripheral nervous systems (*Folts*

*et al., 2019*). In addition, several lines of evidence suggest key functions for aGPCRs in cancer where they may, for example, relay cellular signaling in response to surrounding mechanical cues (reviewed in *Scholz, 2018*).

Mechanosensory roles may prove to be a general feature of these receptors, in relation to the overall structure of aGPCRs and the mode of activation that was recently identified (*Langenhan, 2019*). In aGPCRs, a large extracellular domain, termed the N-terminal fragment (NTF), is linked to the transmembrane 7 domain (7TM), which is referred to as the C-terminal fragment (CTF) of the protein. The NTF and CTF fragments are linked by GPCR autoproteolysis-inducing (GAIN) domain containing an autocatalytic cleavage site (GPS) motif that is activated during receptor processing. Following proteolytic cleavage, the NTF and CTF remain associated via non-covalent interactions during trafficking to the cytoplasmic membrane (*Araç et al., 2012*). Groundbreaking work by *Liebscher et al., 2014* and *Stoveken et al., 2015* has shown that the CTF, in the absence of the NTF, possesses basal signaling activity and that the 15–27 most N-terminal amino acids of the CTF act as a tethered activating ligand, reminiscent of the activation mode of the thrombin receptor (*Seeley et al., 2003*). In light of these data, it is conceivable that interactions of the NTF with extracellular matrix (ECM) components, or with membrane components of neighboring cells, serve as an anchor for the receptor and that binding of an additional ligand, or generation of tension (e.g., shear stress, stiffness of the ECM), changes the structural conformation, thereby releasing or altering the conformation of the NTF, resulting in exposure of the tethered agonist and activation of the receptor.

Several studies have shown constitutive basal activity of CTF constructs and activation of full-length aGPCRs with exogenous synthetic peptides (which we refer to as GPCR-activating peptides [GAPs]) corresponding to their cognate tethered agonist sequence (*Liebscher et al., 2014*; *Demberg et al., 2015*; *Müller et al., 2015*; *Stoveken et al., 2015*; *Wilde et al., 2016*; *Brown et al., 2017*; reviewed in *Bassilana et al., 2019*). Specifically, release of the NTF concomitant to receptor activation has been shown for GPR56/ADGRG1 upon binding to the ECM laminin 111 and transglutaminase 2 (*Luo et al., 2014*) and for EMR2/ADGRE2 bound to the ECM dermatan sulfate and subjected to mechanical cues (vibration) (*Boyden et al., 2016*; *Le et al., 2019*; *Naranjo et al., 2020*). However, it is important to note that some adhesion GPCRs such as CELSR1/ADGRC1, GPR115/ADGRF4, and GPR111/ADGRF2 lack a consensus GPS site and do not show evidence of cleavage. Further, the biological functions of GPR114/ADGRG5 and LAT1 (LPHN1/ADGRL1) are independent of cleavage at the GPS site, suggesting that these receptors may be activated by other structural changes (*Langenhan, 2019*; *Scholz et al., 2017*; *Beliu et al., 2021*).

In order to gain insight into the mechanisms of receptor activation at the level of the TM domains, several studies analyzed activation motifs and conserved functional elements within the TMs from various GPCR classes (*Peeters et al., 2016*; *Arimont et al., 2019*). For example, *Nazarko et al., 2018* performed a comprehensive mutagenesis screen of LPHN1/ADGRL1. For the tethered agonist peptide, a beta strand conformation was suggested followed by a turn element that may be key in receptor activation (detailed in *Vizurraga et al., 2020*). However, only recently a first study explored in greater detail the modalities of the tethered agonist-mediated activation for GPR64/ADGRG2 (*Sun et al., 2021*). This study highlighted the importance of two residues within extracellular loop (ECL)2 and the tryptophan (W) toggle switch in TM6, showing reduced binding of an optimized peptide upon mutation. These functional data were recently complemented by cryo-EM structures of several aGPCRs in the presence of their activating tethered peptide (*Qu et al., 2022*; *Xiao et al., 2022*; *Ping et al., 2022*; *Barros-Álvarez et al., 2022*).

In this study, we set out to investigate the molecular mechanisms of tethered agonist-induced activation of the aGPCR GPR116/ADGRF5. GPR116 is expressed in several tissues and cell types, including alveolar type II epithelial (AT2) cells in the lung. One of the primary roles of GPR116 is in the regulation of pulmonary surfactant as revealed by global knock-out of the gene in mice (*Bridges et al., 2013*; *Yang et al., 2013*; *Fukuzawa et al., 2013*; *Niaudet et al., 2015*). Cell-specific deletion of GPR116 in AT2 cells of adult mice, or deletion of the downstream signaling proteins GNAQ/GNA11, recapitulated the increased pulmonary surfactant phenotype observed in the global knock-out mouse model (*Brown et al., 2017*). With respect to putative ligands, GPR116 has been proposed to associate with surfactant protein D (SFTPD) via its NTF (*Fukuzawa et al., 2013*). However, evidence supporting a direct role for SFTPD in the activation of GPR116 is lacking. Additionally, FNDC4 has recently been shown to activate GPR116-mediated Gs signaling in adipocytes (*Georgiadi et al.,*

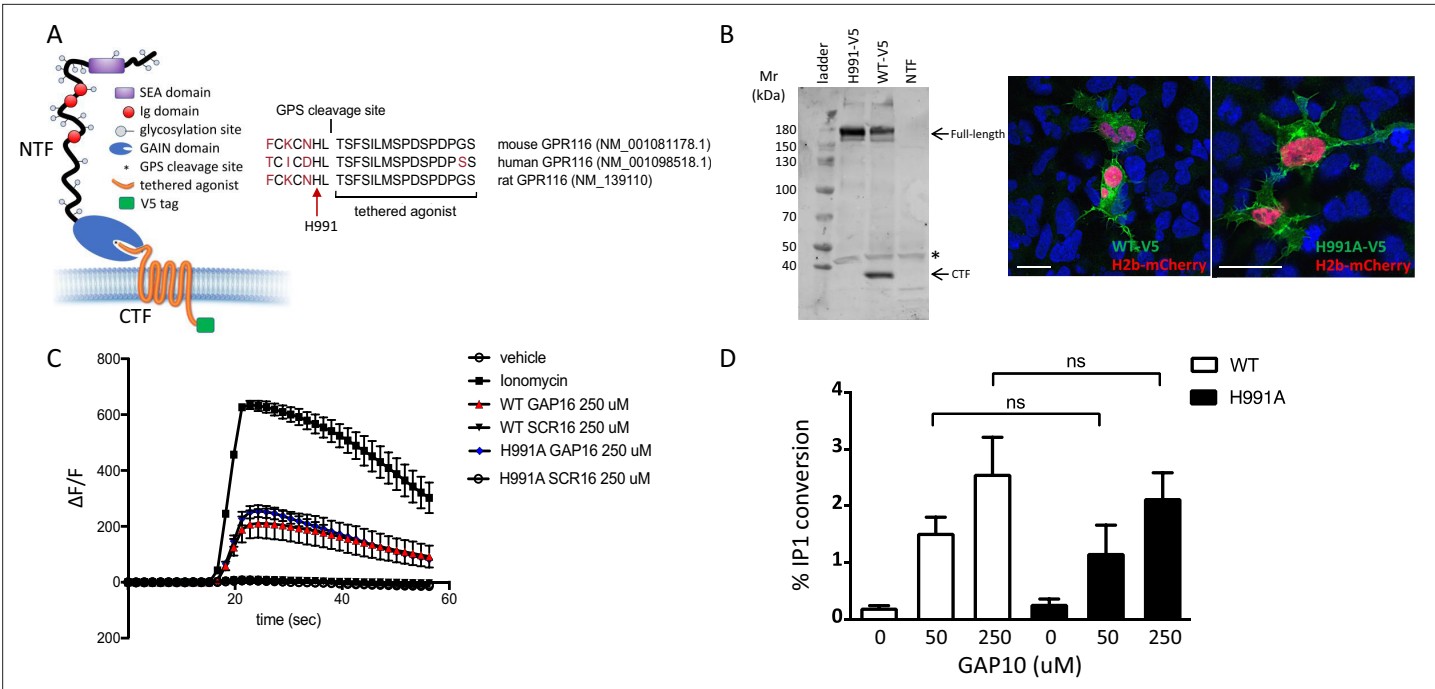

**Figure 1.** Generation and validation of a non-cleavable GPR116 mutant H991A. (**A**) Schematic representation of the H991A mutation in GPR116, details of the sequence at the protein level and gene references. (**B**) Transient expression of V5-tagged H991A in HEK293 cells shows no cleavage at the GPS site by Western blot compared to the wild-type receptor; membrane localization by V5 immunocytochemistry shows similar localization of WT-V5 and H991A-V5. NTF, non-transfected control. Asterisk denotes non-specific band. Scale bars = 25 μm. See also *Figure 1—source data 1*. (**C, D**) Functional characterization of mouse GPR116 H991A in calcium transient assays (**C**) and in IP accumulation assays (**D**), using GAP16, GAP10, or scrambled peptide (SCR16) as the stimulus (n = 3 independent experiments performed with n = 1 technical replicate for **C** and n = 2 technical replicates per group for **D**). Data are expressed as mean ± SD (one-way ANOVA for **C** and **D**). ns, not significant.

The online version of this article includes the following source data for figure 1:

**Source data 1.** Transient expression of V5-tagged H991A in HEK293 cells analyzed by Western blot; original image.

**Source data 2.** Transient expression of V5-tagged H991A in HEK293 cells analyzed by Western blot; uncropped image.

*2021*). Whether FNDC4 modulates Gq-mediated signaling via GPR116 in pulmonary AT2 cells remains to be determined.

In order to evaluate the role of the GPS cleavage and the tethered agonist for in vivo activation and functionality of GPR116, we generated knock-in mice in which autocatalytic cleavage at the GPS site is abolished. This approach has been employed for other aGPCRs in *Caenorhabditis elegans* and in vitro systems, using mutations immediately upstream or downstream of the GPS cleavage site (*Kishore et al., 2016*; *Prömel et al., 2012*; *Peeters et al., 2015*). In addition, we generated synthetic peptides corresponding to the tethered ligand sequence and mutations within the extracellular loops (ECLs) of GPR116 to identify key amino acid residues involved in receptor activation. Finally, we developed a GPR116 homology model based on recently published cryo-EM structures of ADGRF1/GPR110 (*Qu et al., 2022*). Together, our findings provide additional insight into the molecular mechanisms underlying the activation of GPR116/ADGRF5.

## Results
### Physiological relevance of GPS cleavage in GPR116

GPR116 contains a conserved GPS cleavage site, and receptor activation by its corresponding tethered agonist has been demonstrated in previous studies in vitro (*Brown et al., 2017*; *Demberg et al., 2017*; *Zaidman et al., 2020*). Here, we set out to investigate the relevance of GPS cleavage on receptor activation and the physiological functions of GPR116 in vivo. Toward this goal, we first generated an H991A construct containing a histidine-to-alanine mutation at amino acid 991 within the GPS

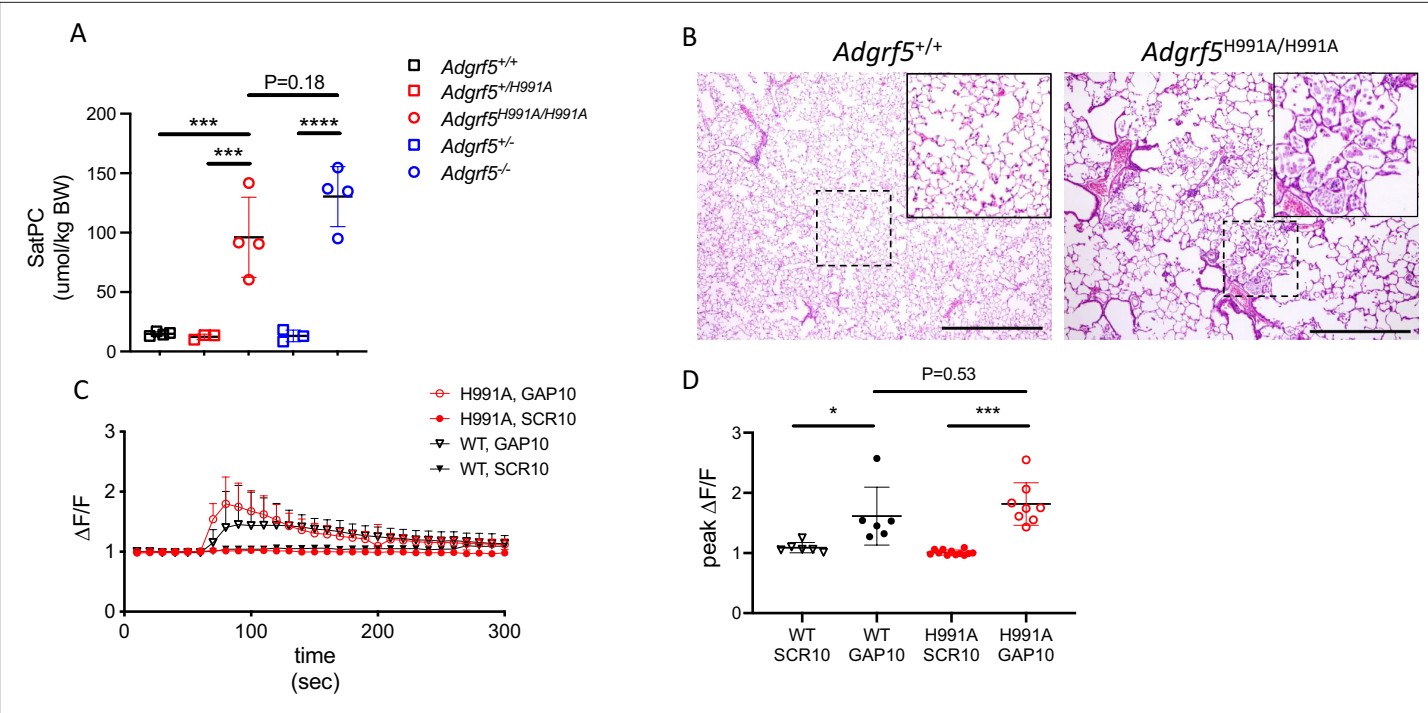

**Figure 2.** Cleavage at the GPS site is required for GPR116 function in vivo. (**A**) Content of saturated phosphatidylcholine (SatPC) in the bronchoalveolar lavage fluid (BALF) of 8-week-old wild-type (GPR116+/+), GPR116+/H991A, GPR116H991A/H991A (line 2552), GPR116+/-, and GPR116-/- mice (n = 3–4 mice per group). Data are expressed as mean ± SD (one-way ANOVA). ***p<0.001, ****p<0.0001. (**B**) Representative histology of 4.5-month-old wild-type and homozygous H991A knock-in mice. Note accumulation of pulmonary surfactant (inset) and alveolar simplification in H991A knock-in mice compared to wild-type control. Scale bars = 500 µm. (**C**) GAP10-induced calcium transients in primary AT2 cells of GPR116+/+ and GPR116H991A/H991A mice (n = 3–4 independent experiments, n = 3 biological replicates per group). Data are expressed as mean ± SD. (**D**) Peak calcium responses in primary GPR116+/+ (WT) and GPR116H991A/H991A AT2 cells treated with SCR10 or GAP10 (n = 3 independent experiments, with n = 2–3 technical replicates per group). Data are expressed as mean ± SD (one-way ANOVA). *p<0.01, ***p<0.001.

The online version of this article includes the following figure supplement(s) for figure 2:

**Figure supplement 1.** Schematic of H991A point mutation introduced into the *Adgrf5* locus via CRISPR/Cas9 gene editing.

of GPR116 (*Figure 1A*). Mutation of the histidine residue at this position, located two amino acids upstream of the cleavage site, was chosen as it is conserved in rodent and human GPR116 and has been shown to abolish autocatalytic cleavage of the GPS in other aGPCRs (*Araç et al., 2012*; *Zhu et al., 2019*). Next, we expressed wild-type and H991A GPR116 in HEK293 cells and demonstrated that the H991A protein is not cleaved relative to the wild-type protein. Specifically, while the CTF fragment and full-length protein were detectable in cells expressing the wild-type construct, only the full-length protein was observed in cells expressing the H991A construct (*Figure 1B*). Although the H991A mutant is not cleaved, this mutation did not impact cellular localization as H991A traffics to the plasma membrane similar to wild-type GPR116 (*Figure 1B*). With respect to signaling, the response of GPR116 H991A to exogenously added agonistic peptide is indistinguishable from that of the wild-type receptor in both calcium transient assays (*Figure 1C*) and in IP conversion assays (*Figure 1D*). Taken together, these results demonstrate that the H991 residue within the GAIN domain is critical for cleavage of GPR116 into NTF and CTF fragments but dispensable for trafficking of the receptor to the plasma membrane and responsiveness to exogenous peptide activation in vitro.

To determine the importance of cleavage at the GPS on GPR116 function in vivo, we generated transgenic mice in which the same single amino acid substitution in the GAIN domain was introduced into the mouse *Adgrf5* locus via CRISPR/Cas9 gene editing (*Figure 2—figure supplement 1A and B*). Bronchoalveolar lavage fluid (BALF) was obtained from three independent lines of 4-week-old H991A homozygous mice and compared to BALF obtained from wild-type mice. Interestingly, all three H991A lines demonstrated increased levels (~65–90 µmol/kg) of saturated phosphatidylcholine (SatPC) compared to BALF obtained from wild-type mice (~10 µmol/kg) (*Figure 2—figure supplement 1C*).

The extent of surfactant overload in H991A mice was comparable to that seen in age-matched *Adgrf5*$^{-/-}$ mice (*Figure 2A*). Histological analysis of homozygous H991A animals revealed accumulation of eosin-positive material and foamy alveolar macrophages in the distal airspaces and alveolar simplification as noted by increased airspace/tissue density (*Figure 2B*), consistent with the pulmonary phenotype observed in *Adgrf5*$^{-/-}$ mice (*Bridges et al., 2013*). To verify functionality of the H991A receptor, we performed calcium transient assays on isolated AT2 cells from homozygous H991A and wild-type mice. These assays demonstrated that H991A AT2 cells respond to exogenous peptide stimulation ex vivo, eliciting calcium transient responses similar in magnitude to that previously observed in wild-type AT2 cells (*Figure 2C and D*; *Brown et al., 2017*). Importantly, expression levels of H991A mRNA were comparable to wild-type GPR116 in primary AT2 cells isolated from H991A and wild-type mice, respectively (*Figure 2—figure supplement 1D*). Taken together, these data demonstrate that while the non-cleavable H991A receptor is fully activated in vitro by exogenous peptides corresponding to the tethered agonist sequence, cleavage of the receptor is critical for GPR116 activation in vivo.

## Identification of key amino acids in the tethered agonist required for GPR116 activation

The functional role of the tethered agonist in GPR116 has been shown previously in vitro using the CTF construct that exhibits strong basal activity due to the presence of the unmasked tethered agonist at the amino terminus (*Brown et al., 2017*). In our previous study, mutation of the tethered agonist amino acids to alanine using sequential steps of three amino acid substitutions for each mutant highlighted the role of this N-terminal sequence in receptor activation. The shortest exogenous peptide capable of activating GPR116 was defined as GAP9, corresponding to the nine most N-terminal amino acids of the tethered agonist. To identify the key amino acids critical for activation, we mutated the 12 most N-terminal amino acids individually to alanine using mouse GPR116 CTF (mCTF) as the parent construct (*Figure 3A*). All mutants showed expression levels comparable to that of mCTF in whole cell lysates, and at the cell surface for key mutations (*Figure 3B and C*). To evaluate the basal activity of the CTF point mutant constructs, we performed IP1 conversion assays and found that three mutants showed complete inhibition of IP1 formation: F995A, L998A, and M999A (*Figure 3D*). Significant effects were also observed for the I997A, D1002, S1003, and P1004 mutations, which showed diminished, but not completely abolished, activity. We then determined whether the inactive mutants would function in a dominant negative fashion or if they could still be activated by exogenous GAP peptides. To this end, we first demonstrated that the wild-type reference mCTF construct could be activated above basal levels (super-activated) with exogenous GAP peptide, and that inclusion of an N-terminal FLAG tag did not affect the response (*Figure 3—figure supplement 1A and B*). Next, we evaluated the different mCTF single alanine mutants for their ability to be activated by the exogenous ligand. In all cases, GAP14 strongly activated the mutant receptors, with IP1 levels accumulating to levels equivalent to or exceeding those observed with the reference wild-type construct (*Figure 3E*). Interestingly, for alanine mutant constructs in which the basal activity was abolished, such as F995A, GAP14 could completely rescue the activity of the mutated tethered agonists. Constructs in which the basal activity was not affected by the mutations were also super-activated by GAP14, similar to the reference mCTF construct. These data are consistent with our previous results with sequential three amino acid substitution mutants (*Brown et al., 2017*) and further show that the tethered agonist single-point mutants are expressed on the cell surface and capable of activation by GAP14, demonstrating that the ligand binding site remains available and accessible in these mutants. Furthermore, these data indicate that F995, L998, and M999 are the key amino acids within the tethered agonist sequence involved in receptor activation.

We next set out to confirm the role of critical amino acids in the tethered ligand using mutated versions of exogenous agonist peptides in the context of the full-length receptor. We used GAP10 as the template for these studies as it is more active on the murine receptor than GAP14. To rule out potential artifacts resulting from oxidation, the methionine corresponding to position 999 in the mCTF was replaced by norleucine (Nle). The activity of GAP10 Nle was indistinguishable from that of parental GAP10 (*Figure 3F*). Calcium flux assays were performed to measure the activity of GAP10 variants (*Figure 3F*). GAP10 peptides corresponding to the mutated CTF constructs F995A, L998A, and M999A failed to activate the full-length receptor, confirming the results shown above using the mCTF constructs with basal activity. Similarly, peptides corresponding to the other singly mutated

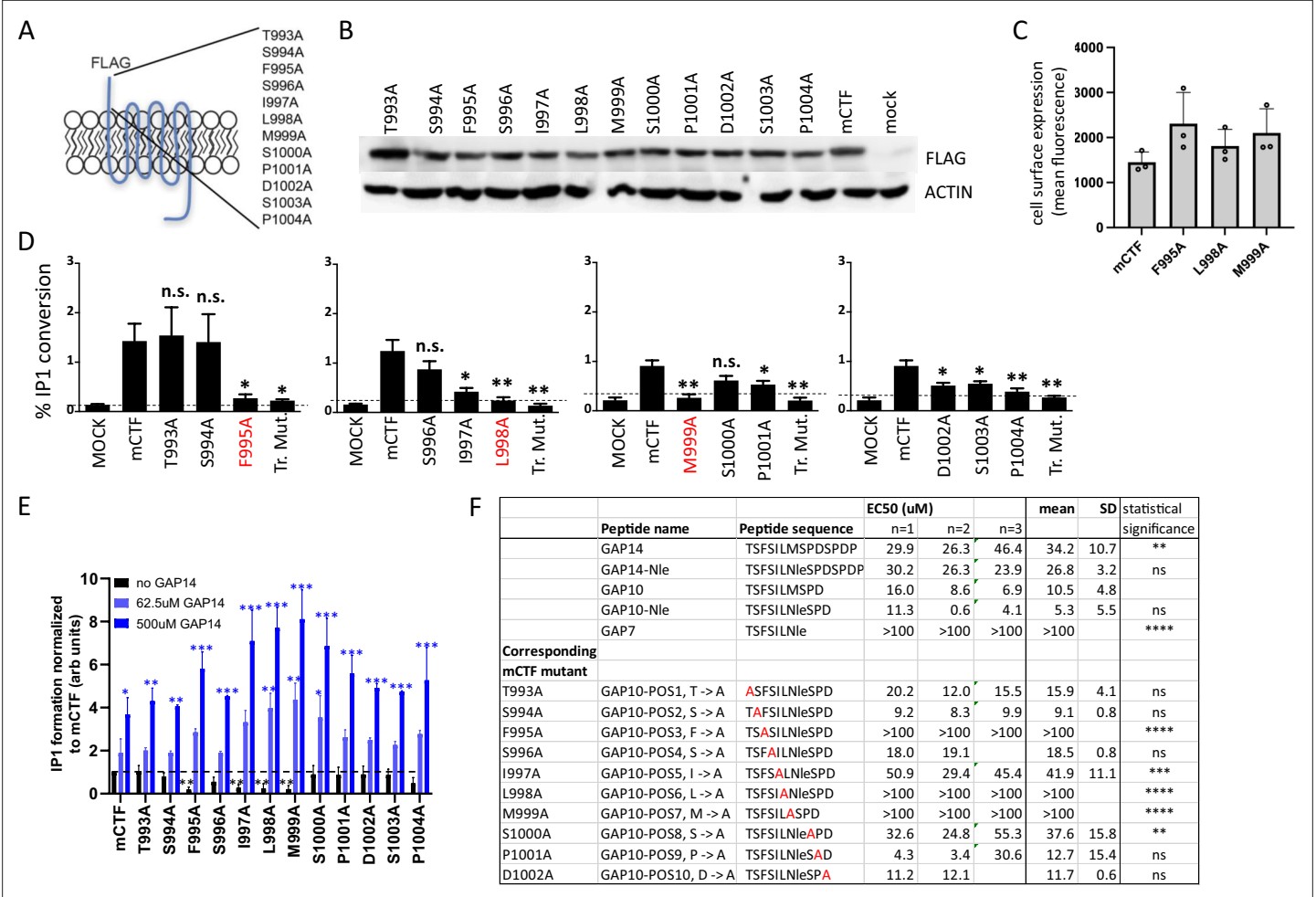

**Figure 3.** Identification of amino acids in the tethered agonist essential for GPR116 activity. (**A**) Design of the 12 mGPR116 CTF (mCTF) ECD mutants. Alanine scan of individual residues in the N-terminal sequence. (**B**) Expression of the mCTF ECD mutants transiently expressed in HEK293 cells. Constructs were detected by Western blot of whole-cell lysates using an anti-FLAG antibody. Moledular weight of FLAG ~43kDa; molecular weight of actin ~42kDa. See also *Figure 3—source data 1 and 2*. (**C**) Quantitation of cell surface expression of mCTF ECD mutants by anti-FLAG antibody staining and flow cytometry of non-permeabilized cells. (**D**) Signaling of the mGPR116 CTF ECD mutants. Constructs of the ECD alanine scan and corresponding triple mutants were transiently expressed in HEK293 cells and basal activity was measured as % IP1 conversion (n = 4–5 independent experiments, with n = 2 technical replicates per group). Dashed line indicates background activity as reference. Data are expressed as mean ± SD. **p<0.01, *p<0.05. The three single-point mutants with the most potent inhibitory effect are highlighted in red. Tr. Mut., triple mutant. (**E**) Exogenous GAP14 treatment rescues mCTF ECD mutants with inactive tethered agonist and super-activates constructs exhibiting basal activity. Constructs were transiently expressed in HEK293 cells and stimulated with GAP14. IP1 accumulation was measured in n = 2–3 independent experiments with n = 2–4 replicates to evaluate basal and GAP14-induced activity of the receptor, respectively. Data are expressed as mean ± SD. Basal activity of mCTF (dashed line) was used as the reference. Basal or GAP14-induced activities significantly different from mCTF baseline are marked with black or blue stars, respectively. *p<0.05, **p<0.01, ***p<0.001. (**F**) Activation of full-length mGPR116 in stable expressing cells (HEK293 clone 3C) with exogenous GAP10 peptides that were sequentially mutated to alanine at each position. Receptor activation was measured via calcium transient assays, in n = 2–3 independent experiments, with n = 4 technical replicates per group. Quadruplicate means of each experiment and final mean ± SD are detailed. **p<0.01, ***p<0.001, ****p<0.0001. See also *Figure 3—source data 3*.

The online version of this article includes the following source data and figure supplement(s) for figure 3:

**Source data 1.** Expression of the mCTF ECD mutants detected by Western blot using an anti-FLAG antibody; original image.

**Source data 2.** Expression of the mCTF ECD mutants detected by Western blot using an anti-FLAG antibody; uncropped image.

**Source data 3.** Activation of FL mGPR116 with exogenous GAP10 peptides that were sequentially mutated to alanine at each position.

**Figure supplement 1.** Super-activation of mGPR116 C-terminal fragment (CTF) constructs with basal activity.

**Figure supplement 2.** Alanine mutation of key amino acids in GAP10 inactivates the peptide for human GPR116 full-length stimulation.

mCTF constructs showed activities comparable or weaker to parental GAP10 in agreement with the results obtained with the mCTF constructs. Importantly, we confirmed that the mutant GAP10 peptides F995A (TSASILNIeSPD), L998A (TSFSIANIeSPD), and M999A (TSFSILASPD) also failed to activate the human full-length receptor, whereas parental GAP10 was active (*Figure 3—figure supplement 2*). These results identify the critical residues of the tethered agonist sequence and further show that functionality of the three most critical amino acids in the GPR116 tethered ligand is conserved across species.

## Identification of key amino acids in the 7TM domain of GPR116 involved in receptor activation

Having identified critical residues in the tethered ligand required for receptor activation and confirmed that GPR116 utilizes the tethered agonist signaling mode, we set out to determine the region of the receptor that interacts with the tethered ligand. We hypothesized that, similar to the PAR1 receptor and as suggested by other aGPCR studies (*Seeley et al., 2003*; *Sun et al., 2021*; *Gad et al., 2021*), critical contact sites for the tethered ligand involve the outside-facing surface of the CTF portion of the receptor, within or proximal to the ECLs.

We therefore performed a comprehensive alanine scan of 51 amino acids located within the three predicted ECLs, including a few amino acids extending into the TMs (*Figure 4A*). Three native alanines located in these regions were mutated to valines. For this scan, we used a mouse GPR116 CTF construct that is deficient in basal activity yet responsive to GAP-mediated activation as the parent construct. This construct, initially described in *Brown et al., 2017*, has the three N-terminal amino acids of the tethered agonist sequence (Thr993, Ser994, F995) mutated to alanines and is referred to hereafter as mCTF 'basal activity deficient' mutant, or mCTFbax. Data from this alanine scan identified five constructs that were unresponsive to GAP14-mediated activation: Y1158A, R1160A, W1165A, L1166A, and T1240A (*Figure 4B*). These constructs exhibited expression levels and patterns comparable to the parent construct, although some had relatively lower detectable surface levels (*Figure 4C and D*, *Figure 4—figure supplement 1A*). Four of these key amino acids are located in ECL2 (Y1158, R1160, W1165, and L1166A) while T1240 is located at the top of TM6, adjacent to ECL3 (depicted as pink circles in *Figure 4A*). Of note, the disulfide bridge between the top of TM3 and ECL2 in GPCRs is known to be required for proper receptor conformation (*Cvicek et al., 2016*). We verified that alanine mutation of C1088, which is predicted to interact with C1164 to form a cysteine bridge, indeed eliminated receptor activation by the agonist peptide (*Figure 4—figure supplement 1B*). All other mutants tested showed comparable activation to the parent construct or partial activation (*Figure 4—figure supplement 1B*; data not shown).

The importance of the five critical ECL residues identified was further validated in corresponding mGPR116 full-length (mFL) constructs, in mCTF constructs with basal activity, as well as in full-length hGPR116 (hFL). Importantly, these residues are conserved between human and mouse (*Figure 4— figure supplement 2*). All five point mutants of mFL were detected at the membrane by immunocytochemistry (*Figure 4—figure supplement 3A*) and showed total expression levels above that of the wild-type receptor (*Figure 4—figure supplement 3B*) but could not, or barely, be activated by exogenous GAP14 (*Figure 4E*). It is noteworthy that the lower expression of mFL is sufficient to result in maximal signaling activity, suggesting that the GPR116 constructs show receptor reserve when overexpressed in HEK cells. Similarly, these five alanine mutations introduced into the hFL receptor abolished the response to GAP14, despite detectable total cell expression levels and localization patterns similar to the hFL receptor (*Figure 4—figure supplement 3C–E*). Mutant T1240A, however, was expressed at lower levels and despite apparent membrane localization this protein lacked functional activity. Replacing these key five residues with alanines in an mCTF construct with high basal activity also strongly decreased or completely abrogated the constitutive basal activity of the mCTF, without affecting expression levels (*Figure 4—figure supplement 4A and B*). Note that the intracellular localization of the C-terminal V5 epitope tag on the mFL and hFL constructs did not permit quantification of cell surface expression of these constructs in these experiments.

In a related series of experiments, we aimed at elucidating whether conservative amino acid replacements within these key residues in mCTFbax would restore receptor activation of the single-point alanine mutants. Introducing phenylalanine instead of alanine for Y1158 indeed allowed for receptor activation, albeit with lower potency for the GAP14 agonist peptide (*Figure 4F*). Mutation of

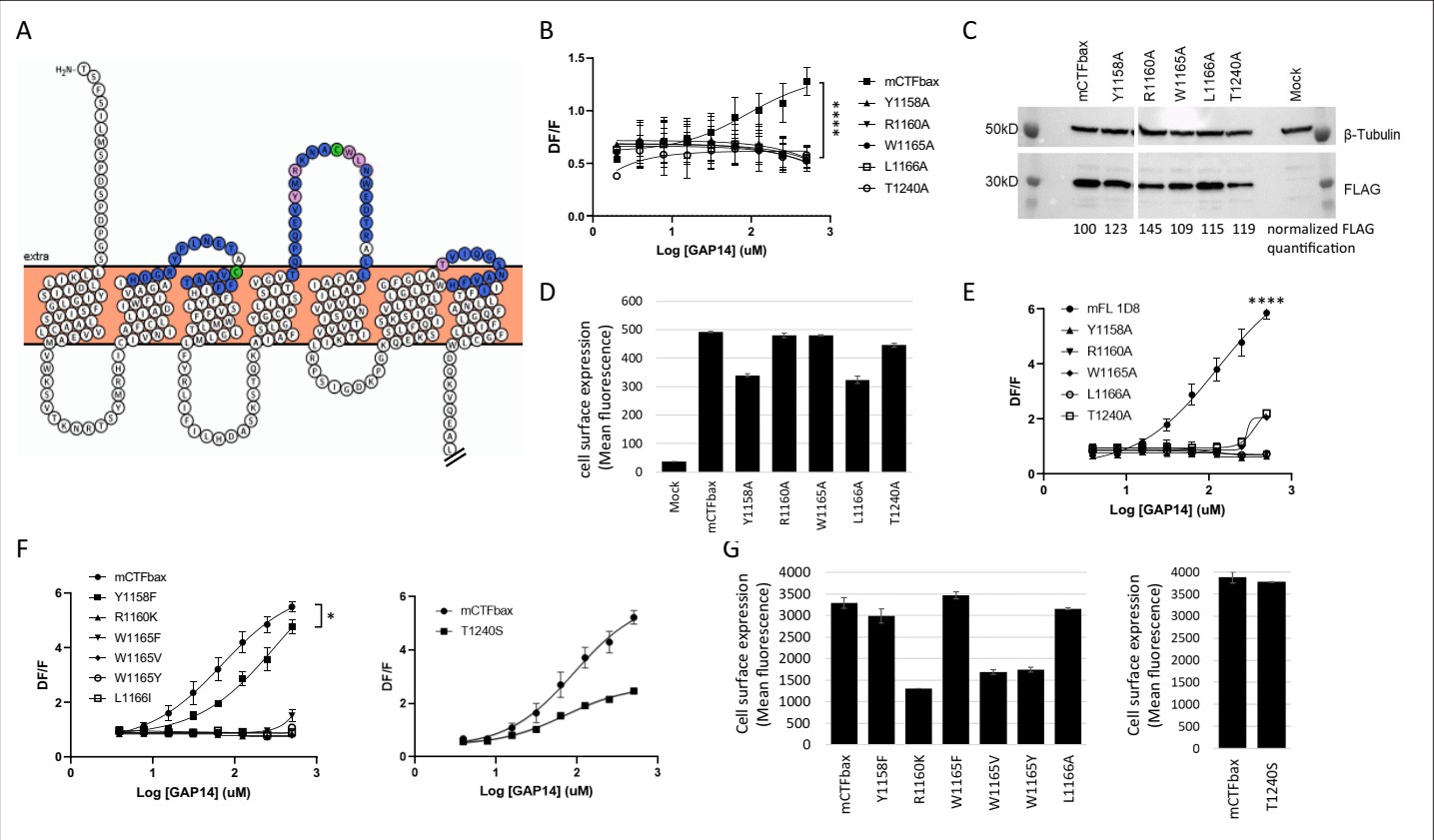

**Figure 4.** Identification of key extracellular loop (ECL) amino acids involved in GPR116 activation by the tethered agonist.
(**A**) Snake plot model of mouse GPR116 C-terminal fragment (CTF). ECL residues individually mutated in the alanine scan of mouse GPR116 CTF constructs tested for functional activity are colored in blue or pink; the five residues shown in pink were identified as strongly modulating receptor activity. The two cysteine residues forming the disulfide bridge are shown in green. The Ct tail was truncated for visualization purposes as indicated by double hash lines. (**B**) Alanine scan of ECLs in mCTFbax, a CTF construct without basal activity. Calcium transient assays were performed in transiently transfected HEK293 cells with exogenous GAP14 as the stimulus. mCTFbax mutants that were completely inactive following GAP14 stimulation are shown. Data are from n = 3 independent experiments, with n = 4 technical replicates per group. Data are expressed as mean ± SD (****p<0.0001 for the reference construct vs. mutants). See *Figure 4—figure supplement 1A* for further alanine scan data, including mutants with no effect. See *Supplementary file 2* for potency and statistical details. (**C, D**) Expression of the mCTFbax mutants not activated by GAP14, in transiently transfected HEK293 cells. (**C**) Proteins were detected by Western blot using an anti-FLAG antibody. For quantification, levels of FLAG signal were normalized to tubulin expression and expressed as % of mCTFbax levels. See also *Figure 4—source data 1–3*. (**D**) Surface levels of receptor expression was measured by flow cytometry using an anti-FLAG antibody on non-permeabilized cells. Data are expressed as mean ± SD from an experiment with duplicates. (**E**) GAP14-induced calcium transients of mFL mutants for five key ECL residues, stably expressed HEK293 cells, as compared to the WT mGPR116 1D8 clone. Data shown are from n = 3 independent experiments with n = 4 technical replicates per group. Data are expressed as mean ± SD (****p<0.0001 for the reference construct vs. mutants). See *Supplementary file 2* for potency and statistical details. (**F**) Mutation of key ECL residues in the mCTFbax construct to amino acids with functional relevance. mCTFbax and mutants thereof were stably expressed in HEK293 cells and GAP14-induced calcium transients were measured. Data are from n = 3 independent experiments, with n = 4 technical replicates per group. Data are expressed as mean ± SD (*p<0.05). See *Supplementary file 2* for potency and statistical details. (**G**) Surface levels of receptor expression for mCTFbax mutants in HEK293 stable populations were measured by flow cytometry using an anti-FLAG antibody. Data are expressed as mean ± SD from an experiment with duplicates.

The online version of this article includes the following source data and figure supplement(s) for figure 4:

**Source data 1.** Expression of the mCTFbax mutants not activated by GAP14, detected by Western blot using an anti-FLAG antibody; original image.

**Source data 2.** Expression of the mCTFbax mutants not activated by GAP14, Western blot of the tubulin control; original image.

**Source data 3.** Expression of the mCTFbax mutants not activated by GAP14, detected by Western blot using an anti-FLAG antibody; uncropped images.

**Source data 4.** Alignment of the human and mouse GPR116 CTF sequences.

**Source data 5.** Expression of hFL WT and mutants, detected by Western blot using an anti-V5 antibody; original image.

**Source data 6.** Expression of hFL WT and mutants, detected by Western blot using an anti-V5 antibody; uncropped image.

**Source data 7.** Expression of mCTFbax mutants, analyzed by Western blot using the FLAG tag; original image.

*Figure 4 continued on next page*

*Figure 4 continued*

**Source data 8.** Expression of mCTFbax mutants; Western blot of the tubulin control; original image.

**Source data 9.** Expression of mCTFbax mutants, analyzed by Western blot using the FLAG tag; uncropped images.

**Figure supplement 1.** Alanine scan of GPR116 C-terminal fragment (CTF) extracellular loops (ECLs).

**Figure supplement 2.** Alignment of the human and mouse GPR116 C-terminal fragment (CTF) sequences.

**Figure supplement 3.** Mutation of the key extracellular loop (ECL) amino acids in mGPR116 full-length (mFL) WT (clone 1D8) and ECL mutants, as well as in human GPR116 full-length leads to inactive constructs.

**Figure supplement 4.** Alanine mutation of key extracellular loop (ECL) residues in mGPR116 C-terminal fragment (CTF) constructs with basal activity and non-conservative mutations in mCTFbax.

T1240 into a serine did not affect the potency of GAP14 activation but led to a decrease in efficacy. Conservative exchanges for the three other critical amino acids still yielded inactive receptors (R1160K, W1165F, W1165V, W1165Y, and L1166I). While Y1158F, W1165V, L1166I, and T1240S showed protein expression levels comparable to the parental mCTFbax construct, R1160K, W1165F, and W1165Y showed a partial reduction in total protein expression levels and membrane localization (*Figure 4G*, *Figure 4—figure supplement 4C*). Despite receptor reserve effect, we cannot exclude the possibility that the absence of activity for these constructs is related to lower receptor expression.

Replacement of W1165 with a phenylalanine, valine, or tyrosine significantly reduced receptor activation. This suggests a role for the -NH group and that the size of the aromatic ring may be relevant. Surprisingly, replacing L1166 with an isoleucine, as is observed in some aGPCRs, was not tolerated in GPR116, suggesting a specific role of this amino acid. Finally, the decreased efficacy of the T1240S mutant suggests that the hydroxyl function is not sufficient to produce a full response to the GAP14-mediated activation. The additional bulk and/or conformational constraint of the methyl group on the T1240 side chain also plays an important role.

Taken together, our data confirms a predominant role for specific amino acids located in ECL2 and at the top of TM6/ECL3 in tethered peptide- and GAP-mediated activation of mouse and human GPR116.

## Species-specific features involved in GPR116 responsiveness to the tethered agonist

The amino acids identified in the CTF that are critical for receptor activation are all conserved between murine and human GPR116 (*Figure 4*, *Figure 4—figure supplements 1–4*); yet GAP peptides consistently elicit a more potent response on the mouse receptor compared to the human receptor (*Brown et al., 2017*), suggesting that additional residues are responsible for this difference in potency. We set out to identify the amino acids relevant for this differential responsiveness to gain additional important insight into the mechanisms underlying GPR116 activation.

We first generated a human GPR116 CTF construct in which the six N-terminal amino acids of the tethered agonist were deleted. This construct, referred to as hGPR116 CTFbax (hCTFbax), has no basal activity, analogous to the mCTFbax construct (*Figure 5A*, inset). As expected, the hCTFbax construct responded to GAP14 activation in IP1 (*Figure 5A*) and calcium flux assays (*Figure 5B*). Similar to observations with the full-length receptors, the hCTFbax is less responsive to GAP14 than the mCTFbax receptor (*Figure 5B*). These constructs were used in subsequent experiments to pinpoint key amino acid residues conferring this differential responsiveness. With this aim we exchanged specific non-conserved amino acids in the human receptor to the corresponding murine residues, focusing on amino acids at the surface of the receptor in the N-terminus and around ECLs (*Figure 5C*). We mutated small clusters of amino acids as summarized in *Figure 5D*, and also introduced the single mutation G1011K, representing a non-conservative amino acid difference between human and mouse sequences in proximity to the extracellular surface. All mutants expressed well at the total protein level and showed localization comparable to that of the parent hCTFbax construct (*Figure 5—figure supplement 1A and B*); note that membrane expression could not be quantified due to cytoplasmic localization of the V5 tag. Calcium flux assays with these constructs demonstrated that, apart from mutant 1242FPGT to IQGS1245, all mutants were activated by GAP14. The 1077QDN to HDG1079 mutant, with the lowest expression of this group of mutants, exhibited an activity close to that of the parent construct, suggesting that the expression levels of these receptor mutants were sufficient

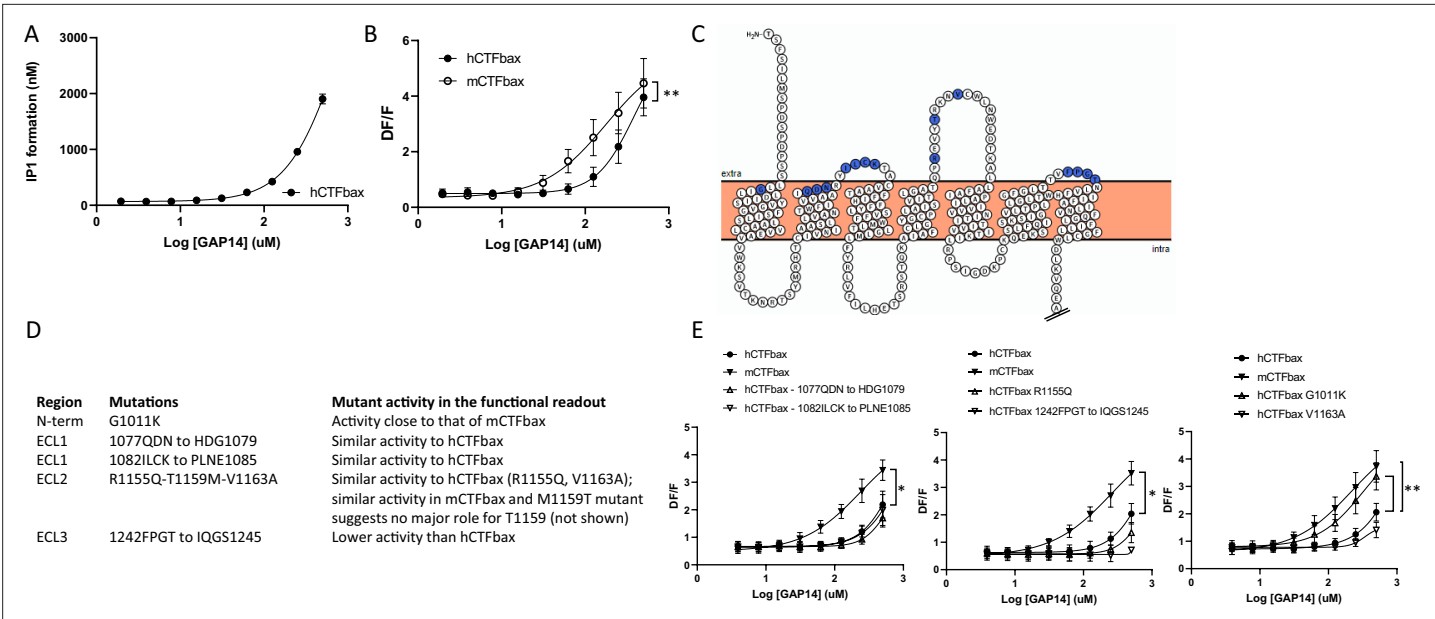

**Figure 5.** Evaluation of the role of extracellular loop (ECL) amino acids not conserved between human and mouse GPR116. (**A**) Characterization of hCTFbax, a C-terminal fragment (CTF) construct without basal activity and that can be activated by GAP14. The hCTF with deletion of the N-terminal six amino acids was stably expressed in HEK293 cells and stimulated with GAP14 in an IP1 accumulation assay (n = 3 independent experiments, with n = 2 technical replicates per group). Data are expressed as mean ±-SD. The basal level of IP1 formation in presence of LiCl is similar to that of parental HEK cells (inset). (**B**) Stable HEK293 populations for mCTFbax and hCTFbax were stimulated with GAP14 and analyzed for induction of calcium transients. Data from n = 3 independent experiments, with n = 2 technical replicates per group. Data are expressed as mean ± SD (**p<0.01). See ***Supplementary file 2*** for potency and statistical details. (**C**) Overview of the mouse-specific residues introduced in the hCTFbax construct. Snake plot of the human GPR116 CTF (including the full tethered agonist sequence); amino acids highlighted in blue, located towards to extracellular side, are not conserved between human and mouse GPR116. The corresponding mouse sequence was introduced in human GPR116. The Ct tail was truncated for visualization purposes as indicated by double hash lines. (**D**) Details of the residues exchanged in the hCTFbax with the corresponding mouse sequence. Each cluster of changes was tested for its signaling capacity and the functional assay outcome is described. (**E**) GAP14-induced calcium transients were not further increased in the mutants as compared to the reference construct hCTFbax with the exception of G1011K (right panel). Constructs were stably expressed in HEK293 cells and tested with n = 2–4 technical replicates (n = 4–5 independent experiments; data are expressed as mean ± SD; *p<0.05, **p<0.01). See ***Supplementary file 2*** for potency and statistical details.

The online version of this article includes the following source data and figure supplement(s) for figure 5:

**Source data 1.** Expression of hCTFbax and mutants thereof, detected by Western blot using an anti-V5 antibody; original image.

**Source data 2.** Expression of hCTFbax and mutants thereof, detected by Western blot using an anti-V5 antibody; uncropped image.

**Figure supplement 1.** Evaluation of the role of extracellular loop (ECL) and N-terminus amino acids not conserved between human and mouse GPR116 C-terminal fragment (CTF).

to generate the maximal signaling response. Surprisingly, only the G1011K mutation increased the signaling response of the resulting receptor so that it shifted sensitivity toward that of mCTFbax (***Figure 5E***). Position 1011 is located at the top of TM1 and immediately downstream of the tethered agonist.

In parallel, while analyzing the potential binding mode of a low molecular weight antagonist specific for the murine receptor discovered in house at Novartis, we began considering additional amino acids deeper in the transmembrane domains of the receptor that might be responsible for the species differences in efficacy. These studies identified a potential binding site containing non-conserved amino acids between mouse and human. We hypothesized that these amino acids might mediate the gain in efficacy for the murine receptor compared to human, as well as inhibition by the antagonist. We identified 10 mouse-specific residues that may be involved in this putative murine-specific binding site (mBS) (***Figure 6A***) and introduced all 10 residues into human GPR116, generating an hCTFbax mBS chimera (***Figure 6—figure supplement 1A***). Compared to hCTFbax mBS, this construct exhibited an enhanced responsiveness to GAP14, similar to that of mCTFbax (***Figure 6B***). We then mutated individually each amino acid of the mBS. Interestingly, introducing the A1254T or V1258A mutation

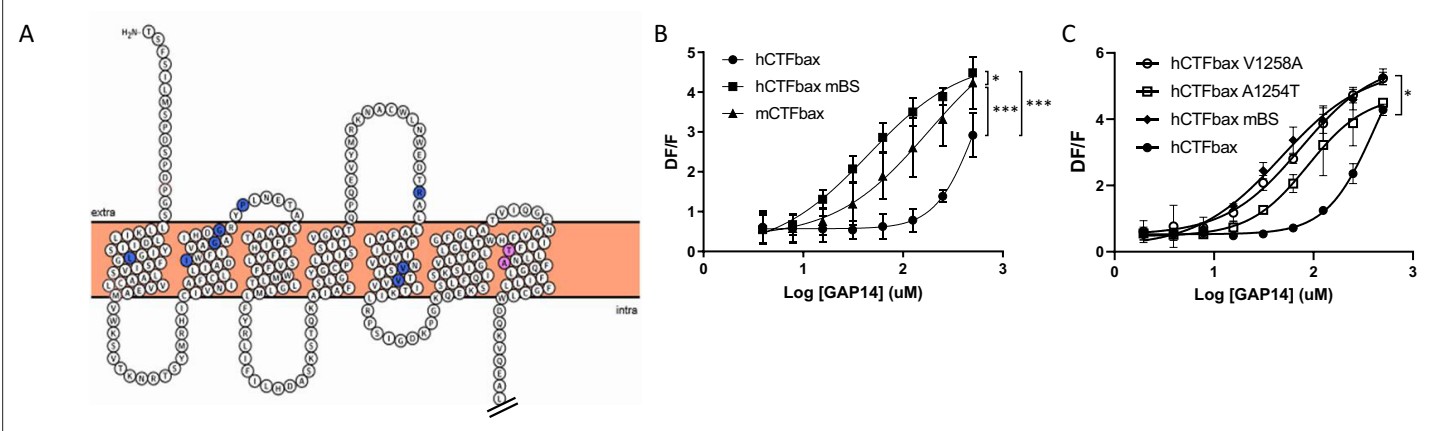

**Figure 6.** Engineering the predicted mouse GPR116 binding site (mBS) into human GPR116 leads to increased peptide activation. (**A**) Snake plot of the mouse GPR116 CTF highlighting mouse-specific amino acids in the putative binding site in blue and pink. Amino acids responsible for mGPR116-specific aspects of receptor activation are shown in pink. The Ct tail was truncated for visualization purposes as indicated by double hash lines. (**B**) Stable HEK293 populations for m/hCTFbax and hCTFbax mBS were stimulated with GAP14 and analyzed for the induction of calcium transients (n = 3 independent experiments, with n = 2 technical replicates per group; data are expressed as mean ± SD; *p<0.05, ***p<0.001). See *Supplementary file 2* for potency and statistical details. (**C**) Characterization of the key amino acids transmitting the effects of the mouse binding site sequence. hCTFbax single mutants were compared to hCTFbax and hCTFbax mBS in calcium transient assays. Stable HEK293 populations were stimulated with GAP14 (n = 3 independent experiments, with n = 2 duplicates per group; data are expressed as mean ± SD; *p<0.05 between hCTFbax and hCTFbax mBS). See *Supplementary file 2* for potency and statistical details.

The online version of this article includes the following source data and figure supplement(s) for figure 6:

**Source data 1.** Expression of hCTFbax and related hCTFbax murine-specific binding site (mBS) A1254T and V1258A mutants; Western blot analysis using an anti-V5 antibody; original image.

**Source data 2.** Expression of hCTFbax and related hCTFbax murine-specific binding site (mBS) A1254T and V1258A mutants; Western blot analysis using an anti-V5 antibody; uncropped image.

**Figure supplement 1.** Details of the putative binding site and expression of corresponding mutants.

into the hCTFbax construct resulted in increased GAP14 responsiveness (*Figure 6C*), similar to the hCTFbax mBS and G1011K constructs. Other non-conserved amino acids in the N-terminus and ECLs appeared to be irrelevant with respect to activation. Expression of these mutants was verified by V5 Western blot and immunocytochemistry (*Figure 6—figure supplement 1B and C*) but the cytoplasmic tag did not permit quantification of membrane expression. Taken together, our data suggest that K1011, T1254, and A1258 in mouse GPR116 are responsible for the increased potency of GAP14 towards the mouse receptor as compared to human GPR116.

## Identification of a mouse-specific activating peptide with enhanced activity

While studying the mechanisms involved in tethered ligand activation of GPR116, we discovered a GAP10 peptide variant with a threonine to proline substitution in the first position (PSFSILMSPD), named GAP10-Pro1, that was at least as potent as GAP10 in activating mFL and mCTFbax, with little to no effect on hGPR116 (FL or CTFbax). Thus GAP10-Pro1 acts as a highly potent, mouse-specific agonist peptide (*Figure 7A*, left panels versus right panels). We also verified that the same pattern was observed with the mFL V5 tagged clone 1D8 (EC50: 107 μM with GAP14, 25 μM with GAP10, 17 μM with GAP10-Pro1). Next, we asked whether the same mouse amino acids that confer increased responsiveness to exogenous GAP14 activation in the human receptor would also enable response to GAP10-Pro1. Indeed, the A1254T and especially V1258A hCTFbax constructs showed a significant response to GAP10-Pro1, albeit not fully equivalent to hCTFbax mBS (*Figure 7B*). This study with a modified peptide confirms that we have identified two important amino acids that determine specific aspects of mouse versus human GPR116 activation. In contrast, the G1011K mutant did not robustly respond to GAP10-Pro1, similar to parental hCTFbax (*Figure 7C*). This latter observation suggests that the proline in the novel peptide interacts with residues distinct from amino acid 1011. In addition

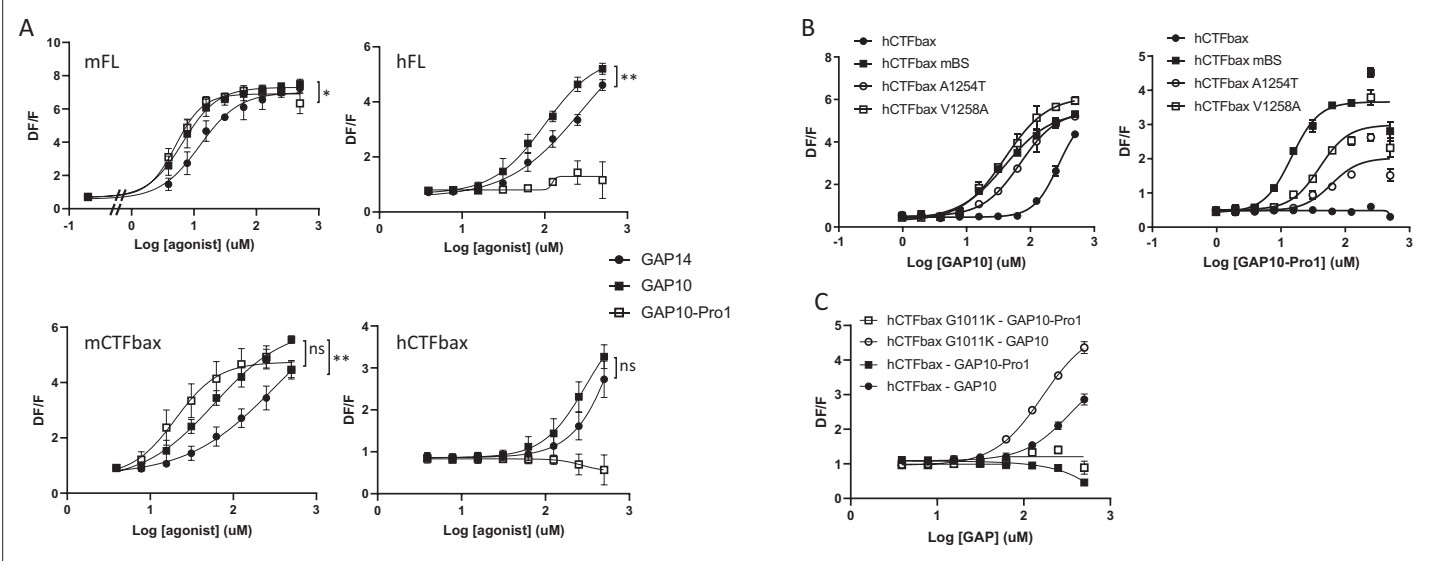

**Figure 7.** Characterization of a mouse-specific agonistic peptide. (**A**) GAP10-Pro1 only activates mouse GPR116. Full-length mouse and human GPR116 (mFL clone 3C and hFL clone A6, respectively), mCTFbax and hCTFbax, all stably expressed in HEK293 cells, were stimulated with GAP14, GAP10, and GAP10-Pro1. Calcium transients are measured as a signaling readout in n = 3–4 independent experiments, with n = 2–4 technical replicates per group. Data are expressed as mean ± SD (*p<0.05, **p<0.01). See *Supplementary file 2* for potency and statistical details. (**B**) Identification of the key amino acids mediating the mouse-specific effects in the mouse binding site. The hCTFbax construct, the hCTFbax mBS chimera, and the single-point mutants A1254T and V1258A in hCTFbax were stimulated with GAP10 or GAP10-Pro1 upon stable expression in HEK293 cells. Activation was measured in calcium transient assays, in n = 2 technical replicates. Data, expressed as mean ± SD, represent n = 4 independent experiments, with n = 2–4 technical replicates per group. See *Supplementary file 2* for potency and statistical details. (**C**) Calcium transients evoked in HEK293 cells stably expressing hCTFbax or the mutated construct G1011K upon stimulation with GAP10 or GAP10-Pro1. Data represent n = 4 independent experiments, with n = 3 technical replicates per group. Data are expressed as mean ± SD. See *Supplementary file 2* for potency and statistical details.

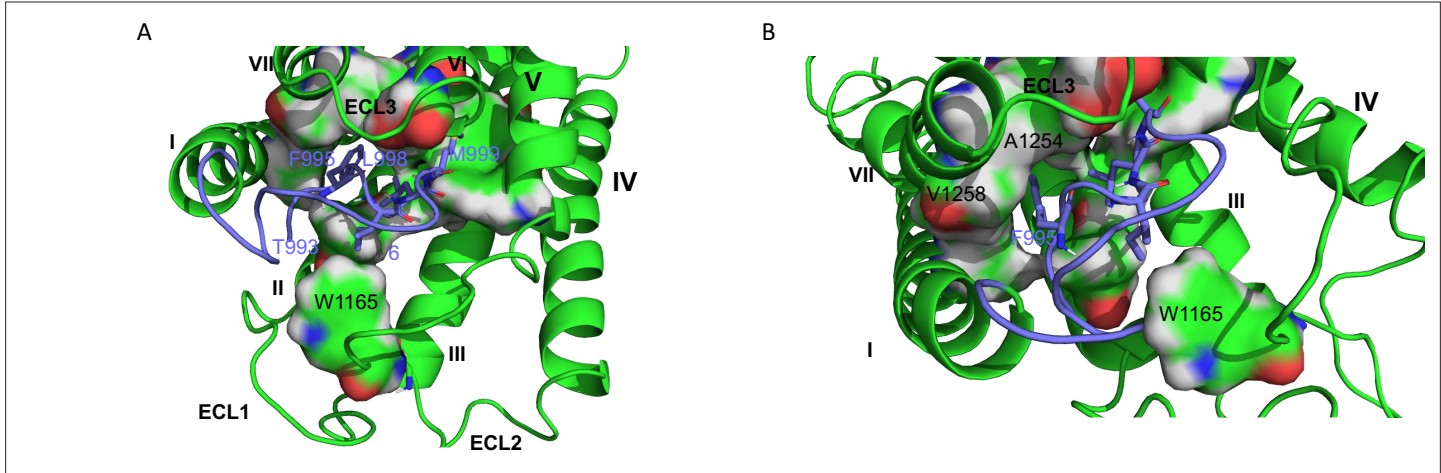

**Figure 8.** hGPR116 7TM homology model. (**A**) Binding model of the tethered agonist peptide (T993-L1010, C atoms in purple) to the transmembrane domain of human GPR116 homology model (C atoms in green). For clarity, only the four amino acids of the tethered agonist peptide that form most of the interactions with the transmembrane domain of GPR116 are displayed as sticks (F995, I997, L977, and M999). The amino acids of GPR116 transmembrane domain that interact with the four amino acids of the tethered agonist peptide mentioned above are displayed as surface color-coded by atom type (C in green, H in gray, N in blue, and O in red). (**B**) Zoom on A1254 and V1258, which form van der Waals contacts with F995.

The online version of this article includes the following source data for figure 8:

**Source data 1.** Homology model of human GPR116 7TM.

to increasing our mechanistic understanding of the mode of tethered agonist function and species differences between human and murine receptors, this newly discovered agonist is an important tool to explore the function of GPR116 in mice in vivo.

## Modeling of human GPR116

In an attempt to build a model for GPR116, we selected the cryo-EM structures of GPR110 in complex with G proteins as a template (*Figure 8A and B*); therefore, one can consider the GPR116 homology model as an active conformation of GPR116.

In this model, the tethered agonist peptide shows the same conformation as in the recent cryo-EM structure of adhesion GPCRs. Its N-terminal half (T993-S1000) binds to a cavity within the helical bundle, while its C-terminal half caps the transmembrane domain. The tethered agonist peptide forms only van der Waals contacts with the transmembrane domain of GPR116. Four hydrophobic side chains of the peptide N-terminal half (F995, I997, L998, and M999, respectively, at positions 3, 5, 6, and 7) are involved in most of these van der Waals contacts (*Figure 8A*). These four hydrophobic side chains, which are well conserved across the ADGRF sub-family of adhesion GPCRs, interact with side chains of amino acids of TM1 (V1022), TM2 (F1070), TM3 (Y1099 and F1103), TM5 (F1177), TM6 (W1234 and T1239), TM7 (F1253, A1254, N1257, and V1259), and ECL2 (W1165).

The proposed interaction model between the tethered agonist peptide and GPR116 is consistent with the identification of GAP9 as the minimal exogenous peptide able of activating GPR116. In addition, the three mutants showing complete inhibition of IP1 formation (F995, L998, and M999; see *Figure 3*) correspond to three of the four side chains that are involved in most of the van der Waals contacts with the TM bundle of GPR116 (F995, L998, and M999). Mutation of the fourth of these side chains (I997) also resulted in a significant inhibition of IP1 formation. Three additional mutations (D1002, S1003, and P1004) resulted in a significant decrease in IP1 formation. In the model, these three amino acids (D1002, S1003, and P1004) do not form extensive interactions with the GPR116 TM bundle. Instead, the side chain of D1002 is involved in electrostatic interaction with the hydroxyl of S1007 (C-terminal half of the tethered agonist peptide), while the hydroxyl of S1003 forms intramolecular H bonds with the side chains of S1000 and D1005. These intramolecular interactions contribute both to stabilize the bound conformation of the tethered agonist peptide and to shield its polar side chains in the hydrophobic environment of GPR116 TM bundle. Therefore, mutation of D1002, S1003, and P1004 is likely to disfavor productive binding of the tethered agonist peptide to GPR116.

Four of the five residues, which when mutated were unresponsive to GAP14 (Y1158, R1160, W1165, and L1166; see *Figure 4*), are located on ECL2, while T1240 is located at the N-terminus of TM6. In the GPR116 homology model, only two of the five key amino acids (W1165 and T1240) interact directly with the tethered agonist peptide, more specifically with I997 and M999, respectively. On the other hand, the homology model does not provide insight into the role of the three other key amino acids on ECL2. However, our modeling protocol may have produced an irrelevant or inaccurate conformation of GPR116 ECL2 because of the challenge associated with modeling long loops. Analyzing the cryo-EM structures of the template may shed some light on the role of the three key ECL2 amino acids. Interestingly, Y1158, R1160, and L1166 are conserved in GPR110 (Y727, R729, and L735). In the three cryo-EM structures of GPR110 (PDB 7WU3, 7WU4, and 7WU5), Y727 and L735 do not form any interaction with the tethered agonist peptide either. However, both amino acids occupy the same region of space. More precisely, Y727 forms van der Waals contacts with TM3, TM4, and ECL2, while L735 is stacked between Y727 and W734 and forms van der Waals contacts with both TM3 and TM5. Therefore, these two amino acids may help to stabilize receptor conformation. On the other hand, the side chain of R729 is more flexible. In one of the three GPR110 structures, it donates an H bond to the backbone O of P575 (amino acid 9 of the GPR110 tethered agonist peptide). Interestingly, this proline is conserved in GPR116 and, as the most C-terminal amino acid of the minimally active peptide GAP9, it may be involved in receptor activation. Finally, the homology model also shows that A1254 and V1258 in TM7 do form contacts with F995 of the tethered agonist peptide (*Figure 8B*), supporting our functional studies in which modification of these residues modulated receptor activity (*Figure 6*).

Taken together, our functional studies combined with molecular modeling provide the first details on the mechanism of GPR116 activation by its tethered agonist peptide.

## Discussion

While great strides have been made in delineating the mechanisms of activation of aGPCRs in vitro, the mode of activation of adhesion GPCRs in vivo, including GPR116, is not well understood. Through generation and characterization of a cleavage-deficient mutant, H991A, we show that cleavage at the GPS site is required for GPR116 activity and modulation of pulmonary surfactant levels in vivo. These studies validate a tethered agonist-mediated activation mode for GPR116 in alveolar type 2 epithelial cells and alveolar homoeostasis. This is, to our knowledge, the first in vivo demonstration of the key role of the GPS cleavage and the unmasking of a tethered agonist in a mammalian system.

The tethered agonist sequence of each aGPCR family member is highly conserved between species, which explains cross-activation among orthologs (*Liebscher et al., 2014*; *Brown et al., 2017*). In this study, we identified F995, L998, and M999 as key amino acids in the tethered agonist sequence required for GPR116 activation. These three amino acids are highly conserved in the ECDs of multiple aGPCR CTFs (*Supplementary file 1*) and have been highlighted for other receptors such as GPR126/ADGRG6, GPR133/ADGRD1, GPR110/ADGRF1, GPR56/ADGRG1 (*Liebscher et al., 2014*; *Stoveken et al., 2015*), suggesting that they play a critical structural role for activation in the aGPCR family. In this sense, these data may explain some of the promiscuous activities observed within and between aGPCR subfamilies (*Demberg et al., 2017*).

Other studies aimed at analyzing the critical residues comprising the tethered agonist are consistent with our identification of phenylalanine at position 3 (F995) as the most N-terminal residue playing a key role in aGPCR activation (GPR64/ADGRG2, *Azimzadeh et al., 2019* and *Sun et al., 2021*; GPR126/ADGRG6, *Liebscher et al., 2014*; Lphn3/ADGRL3, *Mathiasen et al., 2020*). Similarly, amino acids at positions 6 and 7 (L998 and M999), albeit not strictly conserved as leucine and methionine residues in all aGPCR tethered ligands, are key for GPR116 activation. It is noteworthy that additional residues within the tethered agonist that were dispensable for GPR116 activity are critical for other aGPCRs, possibly explaining the specificity for activation exhibited by some receptors. For example, alanine substitution of the first amino acid within the tethered agonist (usually a threonine) can lead to a loss of activity (*Liebscher et al., 2014*) while substitution with a hydrophobic residue can increase the activity of the peptide (*Sun et al., 2021*). This is consistent with our observations with the GAP10-Pro1 peptide, further supporting the concept that position 1 can modulate the strength of molecular interactions between ligand and its binding domain during aGPCR activation.

With respect to the 7TM, our mutational analysis of the extracellular surface of GPR116 identified ECL2 and top of TM6/ECL3 as the structural components most involved in receptor activation. Mutation of key residues may prevent GPR116 from transitioning to an active state or may prevent interaction with the tethered agonist peptide. We identified Y1158, R1160, W1165, and L1166 in ECL2 and T1240 at the TM6/ECL3 interface to be the key residues involved in GPR116 activation. Interestingly, we noted that W1165 and L1166, which immediately follow the cysteine involved in the disulfide bridge linking the top of TM3 and ECL2, comprise a CWL triad that is key for receptor activation. The tryptophan residue is globally conserved in aGPCRs, and the leucine (or sometimes isoleucine) is frequently present in this position (*Supplementary file 1*). Residues within ECL2 have also been implicated in ligand binding and activation of classical GPCRs (*Wheatley et al., 2012*). In particular, the CW motif in the ECL2 is conserved in family B, suggesting commonalities in the activation mechanism between aGPCRs and family B GPCRs. In particular, the (C)WF in ECL2 is involved in CRF1 binding to CRF1R (*Gkountelias et al., 2009*). In PAR1, a (C)HD sequence is important for receptor activation by the peptide, further highlighting the relevance of the ECL2 amino acids following the disulfide bridge, not only for activation per se but also ligand binding (*OBrien et al., 2001*). Within the aGPCR family, *Sun et al., 2021* and *Gad et al., 2021* also identified the (C)WI triad in ECL2 and residues in TM6 as being involved in GPR64/ADGRG2 activation by the tethered agonistic peptide. The tyrosine at position 1158 is located six amino acids upstream of the CWL triad and it is conserved in most aGPCRs, except in the ADGRA and ADGRG subfamily members, and ADGRV1 (*Supplementary file 1*). It is noteworthy that the CWL cluster is also less strictly conserved in these ADGRs. It is possible that a tyrosine at a position further upstream in the ECL2 may functionally replace Y1158 in these receptors, depending on the flexibility of the ECL. With regard to position 1240, a role for ECL3 in aGPCR activation has already been suggested for disease-associated mutations of GPR56/ADGRG1 (*Chiang et al., 2017*; *Kishore and Hall, 2017*; *Luo et al., 2014*). The mutation L640R (corresponding to H1250 in GPR116) in ECL3 affects collagen 3-mediated activation of GPR56 signaling, likely by inducing a

conformation that locks the receptor in an inactive state. While the mutational work presented here highlighted features in line with previously published studies on aGPCR activation, interpretation of the functional data is limited and addressing the role of the identified key amino acids in GPR116 activation was further investigated in a homology model.

Indeed, recently, nine cryo-EM structures of aGPCRs, not including GPR116, were published (*Barros-Álvarez et al., 2022*; *Ping et al., 2022*; *Qu et al., 2022*; *Xiao et al., 2022*). These structures highlight, among other aspects, that the stachel is indeed buried into the 7TM domain, consistent with our data, with a key role of hydrophobic amino acids in positions 3, 4, 5, 7 of the tethered agonist and that the ECL2 can also bend into the barrel with variations depending on the receptor. The agonist peptide is surrounded by ECL2 and TMs on one side, and ECL3 and TMs on the other side. These structures confirm the key role ECL2, and it will be interesting to compare aGPCR activation with that of GPR52, recently shown to involve residues within ECL2 (*Lin et al., 2020*). Although the tryptophan following the cysteine bridge is critical for receptor activation in all aGPCRs, additional essential residues appear distinct in the various studied receptors.

The homology model of human GPR116 confirms that amino acids in ECL2 are important for receptor activation, even if it is surprising that only three residues facing the extracellular side (two in ECL2, one in ECL3) are involved in interactions with the tethered peptide. Based on this model, we could address a role for amino acids identified as key for activation, with some residues being involved in interactions mediating the activation and others being involved in receptor or peptide conformation and stabilization. In particular, the C-terminal part of the peptide (S1000 to S1008) forms intramolecular interactions stabilizing the peptide and Y1158-L1166 in ECL2 may participate in receptor stabilization than activation.

Relative to receptor activation, we identified key residues conserved in the ADGR family such as the tryptophan following the ECL2 cysteine involved in the disulfide bridge or the methionine conserved in most family members at position 7 in the tethered peptide. The homology model also highlighted GPR116-specific residues and interactions, such as R1160 and P1001 conserved in ADGRFs. Further, the species comparison studies highlighted that the mouse sequence-specific amino acids T1254 and A1258 in TM7 strengthen receptor activation by the peptide. The GPR116 homology model shows that the corresponding amino acids in the human sequence, A1254 and V1258, do form contacts with F995 of the tethered agonist peptide. This supports the possibility that the mouse variants induce stronger activation by the peptide, for example, via H bonds by the threonine. Further, it supports our finding that exchanging F995 with proline generates a mouse-specific peptide that is active on a human GPR116 construct exhibiting the mouse sequence at these two positions. The details of such interactions, including the glycine to lysine exchange at position 1011, cannot be predicted using this human GPR116 model and will require further studies.

While the model of the GPR116 activation that we present here is in line with other studies of aGPCRs and highlights aspects of tethered agonist:ECL2 interactions, it does have some limitations. First, further studies using cross-linking and binding experiments will be required to define the contact points between the tethered peptide and ECL2. Second, the potential role of the NTF, how it is positioned relative to the CTF, and how it may affect tethered agonist structure is not known. Modeling the highly flexible loops remains challenging, although some studies focused on modeling of GPCR ECLs are underway (*Wink et al., 2019*; *Liessmann et al., 2021*). In particular, ECL2 is the largest ECL in GPR116, as in many other GPCRs, and the disulfide bridge within TM3 does not introduce sufficient constraints to provide a more defined model. Even with adequate representation of the amino acid stereochemistry, significant degrees of freedom remain for the ECL conformation. Third, our studies evaluate the interaction of the tethered agonist with ECL2 in a context devoid of the NTF. In a setting of peptide- or antibody-based activation without NTF removal, exemplified recently by *Mitgau et al., 2022* for GPR126, these tethered agonist/ECL interactions may be different.

Finally, it will be interesting to investigate whether the proposed FNCD4-mediated activation of GPR116, via binding to the NTF and subsequent G protein-coupling, also involves the tethered peptide sequence or another mechanism to induce activating structural changes in the 7TM.

Based on this study, our working model postulates that following unveiling of the tethered agonist, residues within this sequence interact with amino acids located in ECL2, ECL3, and TM7 to induce conformational changes, resulting in productive Gq/G11 engagement and activation. While the endogenous ligand or the biological process(es) leading to displacement and/or conformational change

of the non-covalently linked N-terminal domain and subsequent GPR116 activation in vivo remain unknown, our data advance the mechanistic understanding of GPR116 activation in the context of pulmonary surfactant homeostasis and may facilitate the development of novel receptor modulators that can be used to treat clinically relevant lung diseases.

## Materials and methods

### H991A knock-in mice

The H991A mutation was introduced into the mouse *Adgrf5* locus via CRISPR/Cas9 editing. The methods for the design of sgRNAs and donor oligo and the production of animals were described previously (*Yuan and Hu, 2017*). Two sgRNAs (g490: GGATGGAGAATGACGTCAGG and g491: TGAG GATGGAGAATGACGTC) were selected according to the on- and off-target scores from the web tool Benchling (http://benchling.com). The selected sgRNA target sequences were cloned into a modified pX458 vector (Addgene #48138) that contains an optimized sgRNA scaffold and a high-fidelity Cas9 (*Chen et al., 2013*; *Slaymaker et al., 2016*). Their editing activity of the sgRNA was validated by the T7E1 assay in mouse mK4 cells (*Valerius et al., 2002*), compared side-by-side with *Tet2* sgRNA that was known to work in mouse embryos efficiently (*Wang et al., 2013*). Validated sgRNA (g490) was transcribed in vitro using the MEGAshorscript T7 kit (Thermo Fisher), purified by the MEGAclear Kit (Thermo Fisher), and stored at –80°C. To prepare the injection mix, we incubated sgRNA and Cas9 protein (Thermo Fisher) at 37°C for 10 min to form ribonucleoproteins and added the single-stranded DNA donor oligo (IDT) to it. The final concentrations were 100 ng/ul sgRNA, 200 ng/ul Cas9 protein, and 100 ng/ul donor oligo. The mutant mice were generated by injection of the mix into the cytoplasm of fertilized eggs of C57BL/6 genetic background using a piezo-driven microinjection technique (*Yang et al., 2014*). Injected eggs were transferred into the oviductal ampula of pseudopregnant CD-1 females on the same day. A total of nine founder pups were born from a single injection, from which tail DNA was analyzed by PCR and sequencing. Three out of nine pups (lines 2552, 2553, 2556) contained the desired H991A mutation on at least one allele and either partial KI, indel or wild-type sequence on the other allele. These three F0 founders were bred with wild-type C57Bl6 mice and F1 offspring were sequenced to identify mice that carried the H991A allele. F1 mice were then bred to establish stable transgenic lines for subsequent analysis. Genotyping was performed with the following primers: wild-type allele: FWD–CCACCTGACGTCATTCTCCA, REV–GGCGCATATAG GAAGTTCGG (product = 188 bp); H991A allele: FWD–CACACAGGCTGTTTCGTTGA, REV–CGC ACTGACTAGTTTCTCCATC (product = 298 bp). Animals were housed in a controlled environment with a 12 hr light/12 hr dark cycle, with free access to water and a standard chow diet. All animal procedures were carried out in accordance with the Institutional Animal Care and Use Committee-approved protocol of Cincinnati Children's Hospital and Medical Center. *Adgrf5*-/- mice were previously described (*Bridges et al., 2013*).

### Constructs

Human and mouse GPR116 full-length constructs (hFL and mFL, respectively) have been described earlier, as well as the mouse CTF constructs (mCTF) with N-terminal mCherry or FLAG (*Brown et al., 2017*). The mCTF mCherry and mCTF (FLAG) constructs exhibit basal activity; the mCTFbax, derived from the latter construct by alanine mutation of the three N-terminal amino acids of the tethered agonist sequence, has no basal activity (*Brown et al., 2017*). Mutants of these constructs have been generated by site-directed mutagenesis using PCR-based techniques.

A human GPR116 CTF construct deleted for the six N-terminal amino acids of the tethered agonist sequence and V5-tagged at its C-terminal end was generated by gene synthesis (including codon optimization; Genewiz, Leipzig, Germany) and cloning into pcDNA3.1; this construct is referred to as hCTFbax and exhibits no basal activity. The hCTFbax construct was generated independently of the mCTFbax, using a strategy (Nt deletion) based on emerging knowledge and appropriate for the planned studies, although no difference in functionality is expected between the two approaches. The hCTFbax mBS chimera was derived from this construct by gene synthesis. Other mutants of the human GPR116 CTF or full-length constructs were generated by site-directed mutagenesis using PCR-based techniques.

The H991A-V5 cDNA construct was generated by gene synthesis (Genewiz, South Plainfield, NJ), sequenced verified and cloned into pcDNA3.1+ for expression in cultured cells. Amino acid numbering of CTF constructs is performed according to the mGPR116 sequence (UniProt G5E8Q8). For hGPR116 CTF constructs, the mGPR116 numbering was utilized as the CTF length is conserved in the two species, leading to a 1 to 1 alignment of amino acids.

## Cells, transfection, and culture

HEK293, HEK293H, and HEK293T cells were obtained from ATCC (Manassas, VA) and used indifferently. HEK293 and HEK293H were grown in DMEM/HAMs F12 medium supplemented with 10% FCS, and 1% penicillin/streptomycin; HEK293T were grown in high glucose DMEM supplemented with 10% FCS, and 1% penicillin/streptomycin. For selection and maintenance of stably transfected cells, G418 (400 µg/ml) was added to the culture medium. Absence of mycoplasma contamination was verified.

Expression constructs were transiently transfected in HEK293 cells using Metafectene-Pro (Biontex T040-0.2; $2 \times 10^6$ cells/well in 6-well plates, transfected with 2 µg plasmid), Lipofectamine 2000 (Invitrogen), PEImax (Polysciences), or the Amaxa Cell Line Nucleofector Kit V (Lonza VCA-1003; $1 \times 10^6$ cells/well in 6-well plates, transfected with 3–4 µg plasmid) according to the manufacturer's protocols. Cells were seeded in the assay plate the next day and the assays were ran 1 day later.

For some constructs, stable populations were generated by treating the cells with G418. After complete selection of resistant cells, these were used in assays. Stable HEK293 clones for full-length hGPR116 (clone A6) and mGPR116 (clone 3C) were described earlier (*Brown et al., 2017*).

## Reagents

All GAP peptides were designed with a norleucine (Nle) replacing the methionine in position 7 of the tethered agonist sequence of GPR116 (e.g., GAP14: TSFSILMSPDSPDP or TSFSILNleSPDSPDP). The Nle modification is shown in *Figure 3E* to have no major impact on the peptide activity and ensures better stability over time. Peptides were ordered at Peptides and Elephants (Hennigsdorf, Germany) or from Thermo Fisher (Waltham, MA) as TFA salts and dissolved as 50 mM stock solution in DMSO or in ddH$_2$O for GAP10 and GAP16 experiments in *Figures 1–3C*. GAP14 as reference peptide for most experiments. GAP10, more active than the minimally active peptide GAP9 (*Brown et al., 2017*), was used for specific studies looking at the role of selective GAP amino acids for receptor activation. Ex vivo and related experiments were performed originally with GAP16 peptide; later, GAP10 was subsequently used as a more active peptide.

## Expression levels of the constructs

### Immunocytochemistry

Expression of the various GPR116 constructs tagged with V5 or FLAG was verified by immunocytochemistry using the Cytofix/Cytoperm kit (BD 51-2091KZ) and following the manufacturer's instructions for fall steps of the process. Briefly, cells were plated in eight-chamber slides (ibidi, Cat# 161107), washed the following day with PBS, fixed and permeabilized for 20 min at 4°C using the BDCytofix/Cytoperm 1× buffer, washed twice, blocked for 20 min at room temperature (RT) (blocking buffer: 1× Wash Solution containing 5% FCS, 1% BSA, 140 mM NaCl, 10 mM HEPES, 5 mM CaCl$_2$) and stained overnight at 4°C using an anti-V5 antibody (1/500; Invitrogen, Cat# R96025) or anti-FLAG antibody (1:100; Sigma, F1365) diluted in blocking buffer, followed by an anti-mouse secondary antibody (V5: 1 hr, RT, 1/500; Invitrogen, Cat# A21200; FLAG: 2 hr, RT, 1:200; Invitrogen, #A11004) according to the manufacturer's instructions. Nuclei were stained for 3 min with Hoechst (Invitrogen, Cat# H-3570). Cells were then analyzed by confocal microscopy on a Zeiss LSM700 inverted microscope.

### Western blot

Transfected cells (approximately $3 \times 10^6$ cells) were plated in a 10 cm or 6-well dish and expression was analyzed 24–48 hr after transfection, in parallel to functional assays. Cells were washed with PBS and lysed in RIPA buffer (Amresco N653-100) complemented with a protease inhibitor cocktail (Calbiochem# 539137). After sonication (40%, cycle 2× for 20 s), total protein concentration of the cell lysates was determined by BCA assay (Pierce) and equal amounts of protein were separated by SDS-PAGE using NuPAGE 4–12% Bis-Tris or Tris-Glycine gels. Proteins were electrophoretically transferred to PVDF membranes (Bio-Rad #1704156) or Amersham Protran nitrocellulose membranes

(Millipore, #GE10600010) and probed with a monoclonal antibody directed against V5 (clone V5-10, Sigma V8012) or FLAG (clone M2, Sigma F1365), and beta tubulin (Cell Signaling, #2128 (9F3)) or actin (Seven Hills Bioreagents, clone C4, Cincinnati, OH) as loading controls. Blots were then probed with donkey anti-mouse/anti-rabbit IgG HRP (Jackson ImmunoResearch 715-035-150/711-035-152) and analyzed for chemiluminescence on a Bio-Rad reader. Quantification of the bands was performed with Image Lab 6.

## Flow cytometry

The expression of mCTF mCherry constructs was quantified by flow cytometry, making use of the N-terminal red fluorescent mCherry (excitation 561 nm, emission 620 nm). Cell surface expression of mCTF FLAG constructs was also quantified by flow cytometry of non-permeabilized cells. For FLAG-tagged constructs, cells were stained with anti-FLAG antibody (Sigma F1804, 1/200) and a PE-conjugated anti-mouse secondary antibody (Invitrogen 12-4015-82, 1:500). Transfected cells ($1\times 10^6$ cells) were analyzed 48 hr after transfection, in parallel with functional assays. Nine thousand cells were analyzed for each construct.

For quantification of membrane expression of the receptor, cells were fixed with 2% PFA for 30 min on ice, washed with FACS buffer (PBS pH 7.2 without calcium and magnesium, 1% BSA, 1 mM EDTA), resuspended in FACS buffer, and blocked for 15 min on ice. After centrifugation, cells were stained for 30 min on ice with an anti-FLAG antibody (Sigma F1804, 1/500), then with an anti-mouse secondary antibody (Invitrogen A21200m 1/500) in the dark. After a final wash in FACS buffer, cells were analyzed with a flow cytometer (Attune or BD FACSCanto). Eight to ten thousand cells were analyzed for each construct.

For quantification of total receptor expression, cells were fixed and permeabilized on ice for 20 min, washed and blocked for 15 min on ice using the Cytofix/Cytoperm kit (BD 51-2091KZ). Staining with the anti-V5 antibody (Invitrogen R96025, 1/500) for 1 hr on ice in the kit Blocking Buffer was followed by washing with the Wash Solution and staining with the anti-mouse secondary antibody (Invitrogen A21200, 1/500) for 20 min on ice and in the dark. After a last wash, cells were resuspended in Wash Solution and analyzed with an Attune flow cytometer. Seventeen thousand cells were analyzed for each construct.

## Calcium transient assays

HEK293 cells were seeded in 384-well plates (Greiner, #781946) at 30,000 cells/well. Medium was removed the following day and cells were loaded for 2 hr at 37°C, 5% $CO_2$ with 20 µl Calcium 6 loading buffer (Molecular Devices, Cat# R8191): 1× HBSS (Gibco, Cat# 14065.049) containing 20 mM HEPES (Gibco, Cat#15630) and 2.5 mM probenecid (Sigma, Cat# P8761). Plates were then transferred to a FDSS7000 fluorescence plate reader (HAMAMATSU, Japan) and $Ca^{2+}$ responses were measured in real time using a CCD camera with excitation 480 nm/emission 540 nm. Prior to stimulation, baseline measurements were recorded over 5 s (Fmin). For stimulation, 20 µl of test compounds prepared 2× in HBSS containing 20 mM HEPES were dispensed. Fluorescence was read over 1–1.5 min. Maximum fluorescence (Fmax) was determined and normalized to baseline. Data are reported as ΔF/F = (Fmax – Fmin)/Fmin. Adequate response of the endogenous M3R was verified for each transfection, using carbachol at 100 µM and following the same procedure as for the peptide activation.

## Calcium imaging of mouse H991A alveolar type 2 cells

Primary mouse AT2 cells were isolated from homozygous H991A adult (mice 6–8 weeks) and cultured in BEGM plus 10% charcoal-stripped FBS and recombinant human KGF (10 ng/ml, PeproTech) on top of a thin substrate of 70% Cultrex (Trevigen)/30% rat tail collagen (Advanced Biomatrix) coated onto 35 mM dishes with a coverslip bottom (MatTek) to maintain a differentiated phenotype. Mouse cells were cultured for 6 days after isolation prior to experimentation. On the day of experimentation, media were replaced with 1 ml Krebs-Ringer's buffer containing 8 mM calcium chloride and incubated for 30 min at 37°C to equilibrate. Cells were washed once with Krebs-Ringer's buffer containing 1.8 mM calcium chloride and loaded with Fluo-4 AM (1 µg/µl; Thermo Fisher) plus 10 µl 0.2% pluronic acid for 30 min at 37°C. Cells were washed three times with 1 ml with Krebs-Ringer's plus calcium buffer, leaving the last 1 ml on cells to image. Baseline fluorescent data were recorded for 5 min prior to addition of SCR10 or GAP10 peptides. Ionomycin (10 µM) was added 2 min prior to end of imaging

as positive control. Time-lapse imaging was performed on Zeiss 200M wide-field fluorescent microscope. Data were analyzed using Slidebook software.

## Inositol phosphate accumulation assays

### Radiometric method

HEK293 cells were seeded in 12-well plates at a density of $1.5 \times 10^5$ cells per well and transiently transfected with plasmids expressing wild-type, wild-type mGPR116-FLAG, CTF-FLAG, or alanine mutants of CTF-FLAG. Then, 24 hr post transfection, media were removed and replaced with serum-free Modified Eagles Medium (MEM; Mediatech) containing 1 µCi/ml of 3H-myo-inositol (PerkinElmer Life Sciences). Also, 48 hr post-infection, media were removed and replaced with serum-free media supplemented with 20 mM LiCl for 3 hr. For samples to be stimulated with GAPs, the peptides were added at the appropriate concentration when the media were replaced with that supplemented with LiCl. Reactions were stopped by aspirating medium, adding 1 ml of 0.4 M perchloric acid, and cooling undisturbed at 4°C for 5 min. Then, 800 µl of supernatant was neutralized with 400 µl of 0.72 M KOH/0.6 M KHCO$_3$ and subjected to centrifugation. Also, 1 ml of supernatant was diluted with 3 ml of distilled H$_2$O and applied to freshly prepared Dowex columns (AG1-X8; Bio-Rad). Columns were washed twice with distilled H$_2$O, total inositol phosphates (IP) were eluted with 4.0 ml of 0.1 M formic acid and 1 M ammonium formate, and eluates containing accumulated inositol phosphates were counted in a liquid scintillation counter. Then, 50 µl of neutralized supernatant was counted in a liquid scintillation counter to measure total incorporated 3H-myo-inositol. Data are expressed as accumulated inositol phosphate over total incorporated 3H-myo-inositol. Adequate response of the endogenous M3R was verified for each transfection.

### HTRF kit

IP1 accumulation was measured using the manufacturer's instructions (IP-One, Cisbio 62IPAPEC). Briefly, cells were seeded at 60,000 cells per well in 384-well plate (Greiner #781080) 1 day before the assay. Plates were flicked and cells were incubated with 20 µl peptide diluted in HBSS buffer (Invitrogen 14065-049) complemented with 20 mM HEPES (Invitrogen 15630-056) and 50 mM LiCl. After 2 hr incubation at 37°C, 5% CO$_2$, 5 µl IP-d2 in lysis buffer and 5 µl IP-Cryptate in lysis buffer were added for 1 hr at RT. Plates were read at 665 and 620 nm on an EnVision Xcite Multilabel reader (PerkinElmer, # 2104-0020). For data analysis, the 665 nm/620 nm ratio was used as data point value and IP1 formation was measured according to the calibration curve.

## Histology of H991A mouse lungs

Adult mouse (4.5 months of age) lungs were inflation fixed with 4% PFA at 20 cm and fixed overnight at 4°C. Tissues were dehydrated the following day and embedded in paraffin. Sections of 7 µm were cut, stained with H&E, and imaged on an X scope.

## Snake plots

Snake plots were generated using the web application Protter – visualize proteoforms (*Omasits et al., 2014*). TM boundaries are as defined in the model.

## Modeling

Positions 972–1270 of the UniProt entry Q8IFZF2 (human GPR116) served as query for a Blast sequence search of the Protein Data Bank (PDB). This search resulted in five hits sharing more than 30% sequence identify with GPR116: PDB entries 7WU3, 7WU4, 7WU5, 7WU2, and 7EPT. The first three entries correspond to the cryo-EM structures of the adhesion GPCR ADGRF1 (GPR110, 51% sequence identity with GPR116), while the last two entries are cryo-EM structures of the adhesion GPCR (ADGRD1 [GPR133, 33% sequence identify with GPR116]). We selected the entries 7WU3, 7WU4, and 7WU5, which belong to the same sub-family of adhesion GPCR as GPR116, as templates for homology modeling. The GPCR domains of PDB entries 3WU3, 3WU4, and 3WU5 are in complex with different heterotrimeric G proteins. Therefore, we extracted the GPCR domain and the tethered agonistic peptide sequences (CTF) of these three PDB entries.

We used Maestro 2022-1 and Pymol 2.3.2 (both from Schrödinger) for all the molecular modeling tasks and picture generation, respectively. To start with, the GPCR domain and tethered agonist

peptide of each entry was submitted to the protein preparation workflow of Maestro using default settings. Then, the 3D structures of the three prepared proteins were aligned. We aligned positions 972–1270 of GPR116 on the sequence alignment derived from the structural alignment. This automatic alignment, performed with default settings, did not require any manual correction. It served as input to build a consensus model of GPR116, derived from each template, using Prime as available from the Maestro 2022-1 interface. More specifically, the homology modeling algorithm placed each Calpha atom at the position where the largest number of templates has that carbon. Then, each of the three intra- and extracellular loops of GPR116 was submitted in turn to the Refine Loops module of Prime. We considered the recommended settings for the loop refinement step. Finally, a run of the Protein Reliability Report module of Maestro 2022-1 on the refined GPR116 homology model did not detect any major geometrical or structural issue.

## Statistical analysis

Concentration response curves were analyzed using GraphPad Prism 9.2.0. Potency was determined using a nonlinear regression model with variable slope, and statistical significance was determined using a one-way ANOVA followed by Dunnett's multiple comparison test or a TTEST for comparison of two data sets (*$p<0.05$, **$p<0.01$, ***$p<0.001$, ****$p<0.0001$, ns, not significant).

## Acknowledgements

We thank the CCHMC gene editing core for their efficient support and Johannes Voshol (Novartis) for helpful discussions throughout this work.

## Additional information

### Competing interests

Caterina Safina, Bernard Pirard, Rochdi Bouhelal, Sejal Patel, Klaus Seuwen, Marie-Gabrielle Ludwig: is employed by and shareholder of Novartis Pharma AG. Nicole Reymann: is employed by and shareholders of Novartis Pharma AG. The other authors declare that no competing interests exist.

### Funding

| Funder | Grant reference number | Author |
|---|---|---|
| National Institutes of Health | HL131634 | James P Bridges |

The funders had no role in study design, data collection and interpretation, or the decision to submit the work for publication.

### Author contributions

James P Bridges, Conceptualization, Resources, Formal analysis, Supervision, Funding acquisition, Validation, Visualization, Project administration, Writing – review and editing; Caterina Safina, Kari Brown, Alyssa Filuta, Ravichandran Panchanathan, Resources, Investigation; Bernard Pirard, Conceptualization, Visualization, Methodology, Writing – review and editing; Rochdi Bouhelal, Conceptualization, Resources, Investigation; Nicole Reymann, Investigation; Sejal Patel, Conceptualization, Resources, Writing – review and editing; Klaus Seuwen, Conceptualization, Resources, Formal analysis, Validation, Investigation, Writing – review and editing; William E Miller, Marie-Gabrielle Ludwig, Conceptualization, Resources, Formal analysis, Supervision, Validation, Investigation, Visualization, Writing – original draft, Project administration, Writing – review and editing

### Author ORCIDs

James P Bridges http://orcid.org/0000-0002-4815-117X
Marie-Gabrielle Ludwig http://orcid.org/0000-0002-7799-4782

### Ethics

All animal procedures were performed under protocols (AS2842_05_22; JPB) approved by the Institutional Animal Care and Use Committee of National Jewish Health in accordance with National Institutes of Health guidelines.

### Decision letter and Author response

Decision letter https://doi.org/10.7554/eLife.69061.sa1
Author response https://doi.org/10.7554/eLife.69061.sa2

## Additional files

### Supplementary files

• Supplementary file 1. Alignment of the human aGPCR tethered agonist and ECL2 sequences. Conserved residues mentioned in the text are highlighted in red and bold. Asterisks indicate aGPCRs for which GPS cleavage has not been shown to date.

• Supplementary file 2. Potency of the various constructs activated by GAPs. EC50s, measured in the calcium transients assay, are given in µM. Number of data points, agonistic peptide used, mean ± SD, and statistical significance are detailed. n.d., not determined; ns, not significant.

• Transparent reporting form

### Data availability

Source data for the modeling is provided (coordinates of the model).

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

# Appendix 1

## Appendix 1—key resources table

| Reagent type (species) or resource | Designation | Source or reference | Identifiers | Additional information |
|---|---|---|---|---|
| Gene (*Mus musculus*) | Adgrf5 | GenBank | NM_001081178.1 | |
| Gene (*Homo sapiens*) | ADGRF5 | GenBank | NM_001098518.1 | UniProt Q8IZF2-1 |
| Genetic reagent (*M. musculus*) | Adgrf5 H991A | This paper | NM_001081178.1 | Germline knock-in of H991A mutation |
| Genetic reagent (*M. musculus*) | Adgrf5-/- | PMID:28570277 | NM_001081178.1 | Germline knock-out |
| Cell line (*H. sapiens*) | HEK293; HEK293T; HEK293H | ATCC; ATCC Gibco | CRL-1573; CRL-3216; 11631-017 | |
| Sequence-based reagent | sgRNA #1 Adgrf5 locus (*M. musculus*) | This paper | | TGAGGATGGAGAATGACGTC |
| Sequence-based reagent | PCR primers Adgrf5 WT allele (*M. musculus*) | This paper | | FWD-CCACCTGACGTCATTCTCCA; REV- GGCGCATATAGGAAGTTCGG |
| Sequence-based reagent | PCR primers Adgrf5 H991A allele (*M. musculus*) | This paper | | FWD-CACACAGGCTGTTTCGTTGA; REV- CGCACTGACTAGTTTCTCCATC |
| Sequence-based reagent | PCR primers site-directed mutagenesis Y1158A (*M. musculus*) | This paper | | FWD-CAGGAAGTCGCCATGAGGAAG; REV-CTTCCTCATGGCGACTTCCTG |
| Sequence-based reagent | PCR primers site-directed mutagenesis R1160A (*M. musculus*) | This paper | | FWD-GTCTACATGGCGAAGAACGCG; REV-CGCGTTCTTCGCCATGTAGAC |
| Sequence-based reagent | PCR primers site-directed mutagenesis W1165A (*M. musculus*) | This paper | | FWD-AACGCGTGTGCGCTCAACTGG; REV-CCAGTTGAGCGCACACGCGTT |
| Sequence-based reagent | PCR primers site-directed mutagenesis L1166A (*M. musculus*) | This paper | | FWD-GCGTGTTGGGCCAACTGGGAG; REV-CTCCCAGTTGGCCCAACACGC |
| Sequence-based reagent | PCR primers site-directed mutagenesis T1240A (*M. musculus*) | This paper | | FWD-GGTCTTGCCGCAGTGATCCAG; REV-CTGGATCACTGCGGCAAGACC |
| Sequence-based reagent | PCR primers site-directed mutagenesis Y1158F (*M. musculus*) | This paper | | FWD-CAGGAAGTCTTCATGAGGAAG; REV-CTTCCTCATGAAGACTTCCTG |
| Sequence-based reagent | PCR primers site-directed mutagenesis R1160K (*M. musculus*) | This paper | | FWD-GTCTACATGAAGAAGAACGCG; REV-CGCGTTCTTCTTCATGTAGAC |
| Sequence-based reagent | PCR primers site-directed mutagenesis W1165Y (*M. musculus*) | This paper | | FWD-GCGTGTTACCTCAACTGGGAGG; REV-CCTCCCAGTTGAGGTAACACGC |
| Sequence-based reagent | PCR primers site-directed mutagenesis W1165V | This paper | | FWD-CGCGTGTGTGCTCAACTGGGAGG; REV-CCTCCCAGTTGAGCACACACGCG |
| Sequence-based reagent | PCR primers site-directed mutagenesis W1165F (*M. musculus*) | This paper | | FWD-AACGCGTGTTTTCTCAACTGG; REV-CCAGTTGAGAAAACACGCGTT |
| Sequence-based reagent | PCR primers site-directed mutagenesis L1166I (*M. musculus*) | This paper | | FWD-GCGTGTTGGATCAACTGGGAG; REV-CTCCCAGTTGATCCAACACGC |
| Sequence-based reagent | PCR primers site-directed mutagenesis Y1158A (*H. sapiens*) | This paper | | FWD-CGGGAAGTCGCTACGAGGAAGA; REV-TCTTCCTCGTAGCGACTTCCCG |
| Sequence-based reagent | PCR primers site-directed mutagenesis R1160A (*H. sapiens*) | This paper | | FWD-GTCTATACGGCGAAGAATGTCTG; REV-CAGACATTCTTCGCCGTATAGAC |
| Sequence-based reagent | PCR primers site-directed mutagenesis W1165A (*H. sapiens*) | This paper | | FWD-GAATGTCTGTGCGCTCAACTGG; REV-CCAGTTGAGCGCACAGACATTC |

*Appendix 1 Continued on next page*

*Appendix 1 Continued*

| Reagent type (species) or resource | Designation | Source or reference | Identifiers | Additional information |
|---|---|---|---|---|
| Sequence-based reagent | PCR primers site-directed mutagenesis L1166A (*H. sapiens*) | This paper | | FWD-GTCTGTTGGGCCAACTGGGAGG; REV-CCTCCCAGTTGGCCCAACAGAC |
| Sequence-based reagent | PCR primers site-directed mutagenesis T1240A (*H. sapiens*) | This paper | | FWD-GGTCTCACCGCTGTGTTCCCAGG; REV-CCTGGGAACACAGCGGTGAGACC |
| Sequence-based reagent | PCR primers site-directed mutagenesis A1254T (*H. sapiens*) | This paper | | FWD-CATCATCTTCACTATCCTGAACG; REV-CGTTCAGGATAGTGAAGATGATG |
| Sequence-based reagent | PCR primers site-directed mutagenesis V1258A (*H. sapiens*) | This paper | | FWD-CCATCCTGAACGCCTTCCAGGGCC; REV-GGCCCTGGAAGGCGTTCAGGATGG |
| Sequence-based reagent | PCR primers site-directed mutagenesis G1011K (*H. sapiens*) | This paper | | FWD-CTCTCTGCTGAAGATCCTGCTGG; REV-CCAGCAGGATCTTCAGCAGAGAG |
| Sequence-based reagent | PCR primers site-directed mutagenesis 1077QDN to HDG1079 (*H. sapiens*) | This paper | | FWD-GCCGCCATCCATGACGGCCGGTACATC; REV-GATGTACCGGCCGTCATGGATGGCGGC |
| Sequence-based reagent | PCR primers site-directed mutagenesis 1082ILCK to PLNE1083 (*H. sapiens*) | This paper | | FWD-AACCGGTACCCCCTGAACGAGACCGCCTGC; REV-GCAGGCGGTCTTGCACAGGATGTACCGGTT |
| Sequence-based reagent | PCR primers site-directed mutagenesis R1153Q- T1157M-V1161A (*H. sapiens*) | This paper | | FWD-CCAGCCCCAGGAAGTCTATATGAGGAAGAATGCGTGTTGGC; REV-GCCAACACGCATTCTTCCTCATATAGACTTCCTGGGGCTGG |
| Sequence-based reagent | PCR primers site-directed mutagenesis 1242FPGT to IQGS1245 (*H. sapiens*) | This paper | | FWD-ACCACCGTGATCCAAGGCTCCAACCTGGTG; REV-CACCAGGTTGGAGCCTTGGATCACGGTGGT |
| Recombinant DNA reagent | H2b-mCherry | Addgene | #20972 | Transfection |
| Peptide, recombinant protein | GAP14 | Peptides and Elephants | PMID:28570277 | TSFSILMSPDSPDP; TSFSILNIeSPDSPDP |
| Peptide, recombinant protein | GAP10; GAP10-Pro1 | Peptides and Elephants or Thermo Fisher | PMID:28570277; this paper | TSFSILNIeSPD; PSFSILNIeSPD |
| Peptide, recombinant protein | GAP16 mouse | Thermo Fisher | PMID:28570277 | TSFSILMSPDSPDPSS |
| Antibody | Anti-V5 mouse monoclonal | Invitrogen | R96025 | Immunocytochemistry; 1/500; Western blot 1/1000 |
| Antibody | Anti-FLAG mouse monoclonal | Sigma | F1365, clone M2 | Immunocytochemistry, 1/100; Western blot, 1/1500 |
| Antibody | Anti-mouse secondary antibody | Invitrogen | A21200 | Immunocytochemistry, 1/500; flow cytometry, 1/500 |
| Antibody | Anti-mouse secondary antibody | Invitrogen | A11004 | Immunocytochemistry, 1/200 |
| Antibody | Anti-V5 mouse monoclonal | Sigma | V8012 | Western blot, 1/1000 |
| Antibody | Anti-beta tubulin rabbit monoclonal | Cell Signaling | 2128, clone 9 F3 | Western blot, 1/1000–1/5000 |
| Antibody | Anti-actin mouse monoclonal | Seven Hills Bioreagents | Clone C4 | Western blot, 1/5000 |
| Antibody | Anti-mouse secondary antibody | Jackson ImmunoResearch | 715-035-150 | Western blot, 1/5000–1/12,000 |
| Antibody | Anti-rabbit secondary antibody | Jackson ImmunoResearch | 711-035-152 | Western blot, 1/12,000–1/15,000 |
| Antibody | Anti-FLAG mouse monoclonal | Sigma | F1804 | Flow cytometry, 1/200; Western blot 1/1500 |

*Appendix 1 Continued on next page*

*Appendix 1 Continued*

| Reagent type (species) or resource | Designation | Source or reference | Identifiers | Additional information |
|---|---|---|---|---|
| Antibody | Anti-mouse secondary antibody | Invitrogen | 12-4015-82 | Flow cytometry, 1/500 |
| Commercial assay or kit | Metafectene-Pro, Lipofectamine 2000, PEI max | Biontex Invitrogen Polysciences | T040-0.2 11668019 24765-100 | Transfection |
| Commercial assay or kit | Amaxa Cell Line Nucleofector Kit V | Lonza | VCA-1003 | Electroporation |
| Commercial assay or kit | Cytofix/Cytoperm | BD | 51-2091KZ | |
| Commercial assay or kit | Calcium 6 | Molecular Devices | R8191 | Calcium transients assay |
| Commercial assay or kit | Fluo-4 AM | Thermo Fisher | F14201 | Calcium transients assay |
| Commercial assay or kit | 3H-myo-inositol | PerkinElmer Life Sciences | NET1177001MC | Radiometric IP assay |
| Commercial assay or kit | IP-One | Cisbio | 62IPAPEC | HTRF IP1 assay |
| Software, algorithm | Maestro 2019-1 | Schrödinger | | Modeling |
| Software, algorithm | Prism 9.2.0 | GraphPad | | Statistics |

