## [Editor Report]

Adhesion GPCRs are a relatively understudied GPCR family. One of the mechanisms they are activated by is a tethered agonist peptide that interacts with the transmembrane domain to activate the receptor. This work first shows that a knock-in mouse expressing a non-cleavable GPR116 mutant to prevent the release of the tethered agonist peptide phenocopies the pulmonary phenotype of GPR116 knock-out mice, demonstrating that tethered agonist-mediated receptor activation is indispensable for GPR116 function in vivo. The study then uses mutagenesis and activity assays to find residues in the tethered agonist that are most important for receptor activation, as well as mutating loops in the extracellular face of the receptor to find residues important in mediating the response to the tethered agonist.

---

## [Decision Letter]

**Decision letter after peer review:**

Thank you for submitting your article "Activation of GPR116/ADGRF5 by its tethered agonist requires key amino acids in extracellular loop 2 of the transmembrane region" for consideration by *eLife*. Your article has been reviewed by 3 peer reviewers, one of whom is a member of our Board of Reviewing Editors, and the evaluation has been overseen by Nancy Carrasco as the Senior Editor. The following individual involved in review of your submission has agreed to reveal their identity: Antony Boucard (Reviewer #3).

Essential revisions:

As detailed in the reviewer comments below, reviewers agree on these essential revisions:

– Cell expression data and quantification for mutants.

– Re-interpretation of the results for the ECL mutations on GPR116 signaling activity.

– Further analysis and re-interpretation of the in vivo mouse experiments.

­– Discussion of previously known mutations about how tethered agonist peptide activates the receptor in introduction, proposed model and discussion.

*Reviewer 1:*

1. Why this abbreviation? "exogenous synthetic peptides (GAP)". It does not fit to the open name.

2. "the cleavage at the GPS site and such receptors may be activated by structural changes not requiring the dissociation of the NTF and CTF." I am not sure if this sentence is needed for the intro.

3. Intro is too long. Authors should condense it into the most important points.

4. Figure 8A. Currently the figure is poorly done. Residues are not labeled well and some residues are shown but not labeled.

5. Figure 1B:

a. The band around 45 kD is a significant band is not mentioned in the text.

b. it is concerning how much intracellular receptor there is and how that may be affecting results (bright green blobs inside cells). Perhaps this is receptor which is internalized after residence at the plasma membrane, or misfolded receptor.

6. Figure 3B: The authors quantify variant expression using whole cell lysates. Yet, if figure 1B is representative, there may be a significant fraction of their GPR116 proteins that are not available on the cell surface, affecting the interpretation of their data in this figure. I would suggest they measure cell surface expression of their proteins and not whole cell lysates. See point 11.

7. Figure 3C: If there is a large amount of internalized/misfolded receptor as seen in Figure 1B, these assay results should be normalized to the level of protein expressed on the surface of the cell, not expression levels from total lysate. See point 11.

8. Figure 3D: not stated how many experiments were performed.

9. Figure 3E: N=1 experiment for IP1 accumulation assay not acceptable. N=2 for calcium assay not preferred. N=3 for both experiments is preferred.

10. In Figure 3E, there are two columns for the calcium EC50. It's not clear which one is the EC50 and what the other column is.

11. Line 235-240: The constructs (N-FLAG/mCherry-CTF-C) described are not detailed in the methods, but referenced. The referenced paper does not describe how long linker regions are. A sufficiently long or flexible linker region between the N-terminal FLAG or mCherry and the tethered agonist could still allow for the tethered agonist to bury within an orthosteric pocket on the aGPCR. Thus, this speculation is not valid, and it actually contradicts their structural model in figure 8A, which shows the tethered agonist binding pocket penetrating deep into the 7TM core of the receptor. Or is this really what is shown in Figure 8? Figure 8 is not clear and needs work with a new model.

12. Line 263- expression levels are claimed to be similar, but they do not look similar from what is shown in Figure 4E. It does not even look like there is any expression for the WT "mFL" protein. They need to quantify it somehow to compare the expression levels. If there are differences in expression their data must be normalized using expression levels.

13. Line 264-265: "… alanine mutations introduced into hFL abolished response to GAP14 while not affecting protein or plasma membrane localization (supp. Figure 6)". However, if you look at supplemental figure 6B, their mutant proteins express much less compared to wild type. In addition, it is not easy to compare membrane expression in figure supp 6C without some method of quantification, but the mutant expression again looks lower than WT. This means their data should be normalized to expression levels, and their new results may actually be different than they are showing in supp. 6A.

14. Line 336: Their expression levels are clearly different from supp. Figure 8. They have to normalize to expression levels.

15. Line 363: "all mutants expressed well": it is difficult to compare without some type of quantification.

16. Line 364/365: It is a stretch to say that hCTFbaxG1011K "resembles mCTFbax". Please reword to something like " the EC50 of hCTFbaxG1011K is shifted towards mCTFbax".

17. Figure 5: Data must be collected in three independent experiments, and it is not acceptable to not show data.

18. Figure 6 and supplemental Figure 10: it is hard to evaluate their claims without quantification of their expression and normalizing their data to their expression.

19. Figure 6B: Data must be collected in three independent experiments. it is not clear how this data was collected. Does "repeat" mean independent experiments?

20. Figure 7 A and B: Data must be collected in three independent experiments

21. Discussion could be trimmed as well.

*Reviewer 2:*

1. To address points 2, 5 and 6 in my public review, the authors should perform additional quantification of data and/or statistical comparisons.

2. To address points 1, 3, 4, and 7 in my public review, the authors should either add further data that supports their claims (if they want to maintain the interpretation of their data as currently written) or alternatively they should alter the discussion of their findings if some of their interpretations cannot be more strongly supported by additional data.

*Reviewer 3:*

1. Title is confusing when referring to "… in extracellular loop 2 of the transmembrane region" and should preferably indicate "… in extracellular loop 2 and the transmembrane region".

2. The authors should analyze the role of the highly conserved Histidine residue in position 991 by generating at least one other substitution mutant with a conservative amino acid. A phenotypic rescue strategy could then be conducted by overexpressing the conservative mutant on the GPR116 knock-out background (previously generated by the research team). A parallel in vitro validation of the receptor mutant in HEK293 cells can be achieved to compare activation by GAP peptides, constitutive activity, expression and cleavage parameters all of which would help reinforce the authors' hypothesis on whether it is a lack of cleavage rather than the substitution of a highly conserved residue that is causing the described phenotypes.

3. Authors should provide a more detailed characterization of the GPR116 H991A/H991A mouse line. First, tissue expression of the mutated receptor should be assessed to discard the probability of a loss-of-function phenotype due to a lack of expression. Second, a direct comparison of AT2 cells secretion parameters should be included for the different genotypes (GPR116+/+, GPR116 -/-, GPR116 H991A/H991A), in order to gain direct mechanistic insights into genotype-linked phenotypes and evaluate the degree of loss-of-function phenotypes between GRP116-deficient mice and cleavage-deficient mice. Third, phospholipid homeostasis parameters should be tested further using AT2 cells from these mice in presence and absence of ATP stimuli (phospholipid secretion, phospholipid uptake).

4. Authors should determine expression levels for each receptor constructs in order to normalize their functional data accordingly. N-terminally Flag-tagged constructs provide an excellent opportunity to evaluate membrane exposure of mCTF receptors from the cell surface of HEK293 cells. This can be achieved by using flow cytometry (as performed in Supp Figure 7 for quantifying N-terminal mCherry signal) quantifying cell-surface fluorescence intensity of cell populations expressing respective receptor mutants which can be detected with anti-Flag antibody and fluorophore-coupled secondary antibody. Alternatively, a cell ELISA can be conducted using the same strategy but revealing with horseradish peroxidase substrates. As for C-terminally V5-tagged constructs, they can be analyzed also by flow cytometry coupled to immunofluorescence in order to obtain total protein expression levels as the positioning of the tag does not allow to make a distinction between subcellular compartments. For western blotting experiments, although band intensities can be visually approximated from western blot images, we highly recommend that authors make use of pixel intensity-based quantification software to monitor band intensities normalized using housekeeping proteins as loading controls.

5. P.11, the authors conducted activation assays rather than binding assays and thus should refrain from inferring that " … set out to determine the region of the receptor to which the tethered agonist binds…". This sentence should be rephrased in order to better reflect the correlation that the authors try to establish using their structure-function assays strategy. Additionally, every mention of a binding site determination throughout the manuscript should be nuanced with this aspect in mind.

6. In their systematic characterization of critical residues by conservative substitutions the authors have dismissed the T1240 position which leaves this seemingly systematic approach incomplete. In order to show consistency, we recommend that they generate a Threonine to Serine replacement at this position and include the analysis in the manuscript.

7. In the generation of basal activity-null constructs ("bax" constructs), it is not clear why a different engineering strategy was adopted to generate the human versus mouse version. The authors should at least allude to it in the manuscript by shortly explaining or indicating if technical or circumstantial elements played a role in designing these different versions, or if other versions of hCTFbax where in fact designed but did not result in reduction in constitutive activity.

8. For consistency purposes, the authors should evaluate all mGPR116 full-length versions of receptor mutants modifying key ECL residues with a version of full-length mGPR116 also containing an intracellular V5 epitope tag to discard any collateral effects due to the insertion of a tag that could potentially disrupt intracellular coupling events. This is particularly important giving that the human versions containing an intracellular V5 epitope (hFL versions) display expression differences (Supp Figure 6B).

9. In order to allow for a more comprehensive reading, the authors should briefly explain in the text what their strategy was when using different GAP peptides interchangeably for the different assays.

10. The manuscript would greatly benefit from the inclusion of a table reporting the EC50 values obtained for all calcium transient experiments as this would allow for a more accurate comparison between the different conditions.

11. Methodology: Immunocytochemistry- Please describe the reagents and their concentrations used for fixation, permeabilization and blocking methods. Calcium transients- Please briefly specify M3R activation protocol.

12. P.15, Please correct the sentence as it contains a repetitive portion: "Conservative exchanges for the three other critical amino acids still yielded inactive receptors… all yielded inactive receptors".

13. P16, please correct "… high probability of interactint…".

14. P.18, please correct "… have been have been…".

15. P.20, please correct "… structural aspects aGPCRs activation, …".

16. Figure 2 panels C and D, please include data obtained for control mice.

17. Figure 2: Please indicate data representation (mean), statistical error and analysis for panels C and D in figure legend.

18. For reproducibility purposes we recommend that the experimental set displayed in Figure 3 panel E for calcium and IP1 accumulation assays, be completed with at least n = 3 and after doing so, please indicate in figure legend the number of experimental repetitions, data/error type and statistical analysis conducted. Also, please indicate in the table the units used for IP1 accumulation assays as well as the respective concentration of indicated activating peptides that were used in this assay.

19. Figure 3 panel D, please indicate in figure legend the proper description of data representation (mean), error and statistical analysis as well as representativity (n of experiments). Also, please include in the panel the "dash line" which is mentioned in the figure legend.

20. Figure 3 panel B, please identify the mCTF lane as such.

21. Figure 8 panel A, please indicate which critical residues are represented by the other spheres at the top of TM3, middle and top of TM2, bottom of TM5, top of TM7 and/or TM2.

22. Please provide scale bars for all microscopy images since some images appear to have been captured with a different zoom magnification (For example Supp. Figure 9).

23. Figure legend for Supp Figure 6, please correct "HEK296" for "HEK293".

24. Figure legend Supp. Figure 10, please indicate if results describe transient or stable HEK293 transfections.

25. Supp. Figure 11, the table's second column title indicates "extracellular part of the NTF…" but should rather say "N-terminal part of the CTF…".

26. Supp. Figure 11, please indicate with an asterisk the GPS cleavage-deficient aGPCR such as ADGRG5.

[Editors' note: further revisions were suggested prior to acceptance, as described below.]

Thank you for resubmitting your work entitled "Activation of GPR116/ADGRF5 by its tethered agonist requires key amino acids in extracellular loop 2 and the transmembrane region" for further consideration by *eLife*. Your revised article has been evaluated by Suzanne Pfeffer (Senior Editor), a Reviewing Editor, and the original reviewers.

We are most intrigued by the work you have reported about the importance of autoproteolysis for the GPR116 function in vivo. Please address the additional issues that the reviewers raised below in their comments including:

– Please discuss the new GPR116 paper that describes a new ligand.

– Clarify the points about reproducibility.

– Revise the title so the story can be better appreciated by the broad audience of *eLife*.

– Rearrange the adhesion GPCR schematic in Figure 1 to correctly indicate the adhesion GPCR domains, including the GAIN domain and the GPS motif which is a peptide inside of the GAIN domain. Please also correct the same issue in the instruction.

– Provide a GPR116 TM/ peptide model based on the new adhesion GPCR structures. The current model is based on a class B family member.

More detailed comments can be found below, and we look forward to receiving your revised manuscript.

*Reviewer #2 (Recommendations for the authors):*

The authors were very responsive to the reviewers' comments and addressed all of the reviewers' concerns. The textual changes and new data added by the authors have significantly strengthened the manuscript.

One additional comment I would like to make is that after the review of the original draft of this manuscript in Spring 2021, a paper was published reporting the secreted protein FNDC4 as a ligand for GPR116 (Georgiadi et al., Nat. Comm., 2021). This work is not mentioned in the present manuscript, and indeed the authors have a statement in the penultimate paragraph of their Introduction stating, "Still considered an orphan aGPCR, GPR116 has been proposed…".

Is GPR116 still an orphan? It is understood that FNDC4 cannot be considered a definitive ligand for GPR116 until other groups have had a chance to confirm (or refute) the findings of Georgiadi and colleagues, but nonetheless, it seems that the work of Georgiadi et al. is highly relevant to the studies described in the present manuscript and therefore should at least be mentioned at some point.

Along these same lines, it is terrific that the authors of the present manuscript made the effort to cite the recent wave of cryo-EM studies on aGPCRs and place their findings in the context of those new structural studies. In a similar vein, it would also be appropriate for the authors to cite the recently-published work of Georgiadi et al. and discuss how these findings may be relevant to the authors' own findings of the activation mechanism of GPR116.

*Reviewer #3 (Recommendations for the authors):*

I am satisfied with the authors' response to my original comments, but I have some suggestions that would need to be addressed for ease of reading, accuracy, and reproducibility purposes regarding the newly added data in this revised version:

1. To avoid confusion, I would suggest adding "Cell surface expression (Mean fluorescence)" as the title for the Y axis of all graphs reporting results from surface exposed epitope in flow cytometry (Flag): Figure 3C, Figure 4 D, Figure 4 Supp figure 4D-E.

2. Likewise, I would suggest indicating "Total expression (mean fluorescence)" as the Y-axis title of all graphs reporting flow cytometry results from intracellular epitopes (V5/mCherry). Figure 4E, Figure 4 Supp Figure 4B.

3. Refrain from using the terms “membrane expression” when the methodology that has been used does not allow this determination exclusively since the immunocytochemistry protocol, as currently described in the methods section, involves a permeabilization step. This is the case in Figure 4 Supp figure 3A and 3C.

4. Please complete the newly added data with the corresponding information on reproducibility parameters (# of independent experiments, define error bars). Error bar descriptions and/or # of independent experiments are missing for various figures involving functional assays or for expression quantifications by western blot or flow cytometry (western blot quantifications do not display errors? Was it done just once? Justify if this is the case as stable cell lines can show decaying expression with time). These include: Figure 3F; Figure 3 Supp Figure 1 and 2; Figure 4; Figure 4 Supp Figure 1B,3B,3D, 4B-E; Figure 5A,B,E; Figure 5 Supp Figure 1B; Figure 6B-C; Figure 6 Supp Figure 1C; Figure 7; Figure 10 (error is SD or SEM?).

5. It seems that flow cytometry experiments were performed only once, indicate if this is not the case.

6. For flow cytometry experiments, indicate the number of events analyzed in the methods section.

7. The use of the term “biological replicates” seems incorrect throughout figure legends as they seem to indicate technical replicates instead. One experiment done at the same time is a n=1 that might include various replicates of each different groups, considered technical replicates; when the same experiment is done more than once at different times experiments are considered biological replicates. For example, the legend for Figure 4F indicates 3 independent experiments with 4 biological replicates but instead should read as 3 independent experiments done in quadruplicates or 4 technical replicates. Please revise where relevant and correct if this is the case or add if it is missing (For example Fig3F, Figure 3 Sup Figure 1, and others are missing replicates data).

8. Indicate where relevant when experiments are technical replicates instead of biological replicates. Add information about statistical significance where data represent biological replicates.

9. Because the data seems more relevant to the reader in this case, please interchange current panel E in Figure 4 with surface expression graphs for remaining mutants displayed in this figure depicted in Figure 4 Supp Figure 4 D-E.

10. Line 660, please correct the antibody incubation time indicating 230 min.

11. Indicate “% IP1 conversion” as the Y-axis title for all corresponding graphs.

12. Note that no cell surface data expression is present for hFL-T1240A and is also missing for various hFL or hCTFbax or mCTF mutants. If data cannot be included at this stage, please include a brief comment in the manuscript to the effect of why this is the case in each section where the data is missing.

13. Figures 9 and 10 seem more suited for the supplementary section.

---

## [Author Response]

Essential revisions:As detailed in the reviewer comments below, reviewers agree on these essential revisions:– Cell expression data and quantification for mutants.

We have addressed these points by adding additional quantification of cell surface expression for the key functional mutants identified in this study, as permitted by the N-terminal or C-terminal epitope tags present in the expression constructs (Figure 3C; Figure 4C,D and E; Figure 4 supplement figure 3D; Figure 4 supplement figure 4C,D and E; Figure 5 supplement figure 1B; Figure 6 supplement figure 1C). We have also added additional statistical analyses for all the dose-response curves (Figure 3E and F, Figure 4B,F and G, Figure 5B and E, Figure 6B and C).

– Re-interpretation of the results for the ECL mutations on GPR116 signaling activity.

We rephrased these aspects as requested in the Results section and discussed their relevance in context of the presented and published data.

– Further analysis and re-interpretation of the in vivo mouse experiments.

We have included additional data demonstrating comparable surfactant lipid accumulation in the bronchoalveolar lavage fluid of H991A and WT mice (Figure 2A), added additional data for ex vivo activation of isolated H991A and WT alveolar epithelial cells (Figure 2C and D) and altered our interpretation of the in vivo experiments in the Results and Discussion sections as requested.

­– Discussion of previously known mutations about how tethered agonist peptide activates the receptor in introduction, proposed model and discussion.

We have included discussion of mechanisms of tethered agonist peptide activation of the receptor from published data of additional aGPCRs in the Introduction, Results and Discussion sections, as requested.

Reviewer 1:1. Why this abbreviation? "exogenous synthetic peptides (GAP)". It does not fit to the open name.

We use the term GAP (GPCR-activating peptide) to refer the numerous synthetic peptides used in this study. We have added an explanation of the GAP abbreviation to the text.

2. "the cleavage at the GPS site and such receptors may be activated by structural changes not requiring the dissociation of the NTF and CTF." I am not sure if this sentence is needed for the intro.

We have changed this sentence in the Introduction.

3. Intro is too long. Authors should condense it into the most important points.

We condensed the Introduction as requested. (1333 words down to 938).

4. Figure 8A. Currently the figure is poorly done. Residues are not labeled well and some residues are shown but not labeled.

We have edited Figure 8 for clarity and added additional data including modeling of GPR116 with the glucagon receptor and its cognate ligand.

5. Figure 1B:a. The band around 45 kD is a significant band is not mentioned in the text.

This is a non-specific band since it is present in all three lanes, including the non-transfected (NTF) control. We have edited the panel to highlight this band with an asterisk and refer to such in the figure legend.

b. it is concerning how much intracellular receptor there is and how that may be affecting results (bright green blobs inside cells). Perhaps this is receptor which is internalized after residence at the plasma membrane, or misfolded receptor.

We consistently observe some degree of intracellular staining in cells transfected with GPR116, more so with the CTF constructs than with the full length receptor, the mechanism for which is presently unknown. However, desensitization experiments did not show decrease of activity of the mFL construct over a 2 hour time frame (not shown). Peptide-induced receptor internalization studies are challenging for this reason as well. That said, the degree of intracellular staining is similar for WT and the H991A mutant, so if signaling were affected by localization within the cell one would predict it would be affected similarly for both constructs. HEK293 cells are a strong expression system for CMV promoter-driven genes and indeed a strong receptor response is observed in functional assays, suggesting high levels of appropriate expression and localization. Further we do know that GPCRs exogenously expressed in HEK293 cells can rapidly reach receptor reserve (i.e. maximal response is reached without complete receptor occupancy – see e.g. the review from Buchwald 2019: Frontiers | A Receptor Model With Binding Affinity, Activation Efficacy, and Signal Amplification Parameters for Complex Fractional Response Versus Occupancy Data | Pharmacology (frontiersin.org)), implicating that lower membrane expression can be sufficient to reach maximal response.

6. Figure 3B: The authors quantify variant expression using whole cell lysates. Yet, if figure 1B is representative, there may be a significant fraction of their GPR116 proteins that are not available on the cell surface, affecting the interpretation of their data in this figure. I would suggest they measure cell surface expression of their proteins and not whole cell lysates. See point 11.

We have quantitated cell surface expression of the key N-term FLAG and ECL variants critical for GPR116 activation using FACS analysis of non-permeabilized FLAG-stained cells (Figure 3C and Figure 4D and E, Figure 4 – supplement figure 4D and E, respectively). These data demonstrate similar levels of cell surface protein expression amongst the wild-type and mutated receptors, except for 3 mutants with conservative amino acid changes which is discussed in the Results section.

7. Figure 3C: If there is a large amount of internalized/misfolded receptor as seen in Figure 1B, these assay results should be normalized to the level of protein expressed on the surface of the cell, not expression levels from total lysate. See point 11.

Expression levels of GPR116 protein present at the cell surface have been performed (Figure 3C). Please see response to inquiry 6 above.

8. Figure 3D: not stated how many experiments were performed.

The graph in the original submission was from a single experiment. We have now added additional data for the basal activity and the GAP14 super-activation along with appropriate statistical analyses.

9. Figure 3E: N=1 experiment for IP1 accumulation assay not acceptable. N=2 for calcium assay not preferred. N=3 for both experiments is preferred.

We have removed the IP1 data and generated additional calcium assay data for a total of n=3 experiments. Two of the peptides (S996A and D1002A) were degraded and could not be re-tested for a triplicate value in this assay.

10. In Figure 3E, there are two columns for the calcium EC50. It's not clear which one is the EC50 and what the other column is.

We have performed an additional replicate for all peptides, except for S996A and D1002A for reasons stated in query 9 above, included statistical analysis and edited the layout of this figure for enhanced clarity.

11. Line 235-240: The constructs (N-FLAG/mCherry-CTF-C) described are not detailed in the methods, but referenced. The referenced paper does not describe how long linker regions are. A sufficiently long or flexible linker region between the N-terminal FLAG or mCherry and the tethered agonist could still allow for the tethered agonist to bury within an orthosteric pocket on the aGPCR. Thus, this speculation is not valid, and it actually contradicts their structural model in figure 8A, which shows the tethered agonist binding pocket penetrating deep into the 7TM core of the receptor. Or is this really what is shown in Figure 8? Figure 8 is not clear and needs work with a new model.

In the tagged constructs, the FLAG or mCherry tags are located immediately downstream of the ATG start site and a prolactin signal sequence, in frame with the coding sequence for GPR116 with no linker regions. We have included this information in the revised Methods section.

In addition, we have performed a structural alignment of a GPR116 homology model with the GLP1 receptor in complex with its ligand glucagon (Figure 8D). This analysis revealed that the N terminus of glucagon overlaps with the largest binding site of GPR116, suggesting that a peptide is capable of binding, at least partially, to the binding site we proposed. We adapted the text accordingly and attempted to make the point with more clarity.

12. Line 263- expression levels are claimed to be similar, but they do not look similar from what is shown in Figure 4E. It does not even look like there is any expression for the WT "mFL" protein. They need to quantify it somehow to compare the expression levels. If there are differences in expression their data must be normalized using expression levels.

The WT mFL clone used in the original figure was not tagged, therefore we have redone all parts of this figure using a V5-tagged mouse FL clone 1D8 as the control (Figure 4E,F and Figure 4 supplement figure 3A). Receptor expression was quantified by V5 flow cytometry and shows higher expression of the mutants than the control reference construct. As this tag is intracellular on the C-terminus, only total expression can be quantified. It is complemented by immunocytochemistry data to demonstrate localization of the protein to the plasma membrane.

13. Line 264-265: "… alanine mutations introduced into hFL abolished response to GAP14 while not affecting protein or plasma membrane localization (supp. Figure 6)". However, if you look at supplemental figure 6B, their mutant proteins express much less compared to wild type. In addition, it is not easy to compare membrane expression in figure supp 6C without some method of quantification, but the mutant expression again looks lower than WT. This means their data should be normalized to expression levels, and their new results may actually be different than they are showing in supp. 6A.

In particular T1240 is indeed expressed at lower levels than the other constructs (see added WB quantification) but expression level seems reasonable (see also ICC).

We have edited the text to dampen our conclusions as requested.

14. Line 336: Their expression levels are clearly different from supp. Figure 8. They have to normalize to expression levels.

We have quantified the Western Blot (Figure 4 supplement figure 4C) and added FLAG flow cytometry data showing the membrane levels of the constructs (Figure 4 supplement figure 4D,E). We edited the text accordingly, in case the 50% reduction in membrane expression would not be sufficient for generating the full activity levels.

15. Line 363: "all mutants expressed well": it is difficult to compare without some type of quantification.

We have added a Western Blot, with quantification, showing the total expression of the constructs to complement the immunocytochemistry data (Figure 5 – supplement figure 1B). Expression of the constructs can be considered as reasonable according to the pattern seen by immunocytochemistry and the least expressed mutants, 1077QDN to HDG1079, with the lowest expression level by Western Blot, still shows WT-like activity in the functional assay.

16. Line 364/365: It is a stretch to say that hCTFbaxG1011K "resembles mCTFbax". Please reword to something like " the EC50 of hCTFbaxG1011K is shifted towards mCTFbax".

We followed the suggestion and adapted the wording accordingly.

17. Figure 5: Data must be collected in three independent experiments, and it is not acceptable to not show data.

We have added data as recommended and the graph now represents a mean of n=3-5 independent experiments (Figure 5E).

18. Figure 6 and supplemental Figure 10: it is hard to evaluate their claims without quantification of their expression and normalizing their data to their expression.

We have included a Western Blot for increased clarity on the expression of the constructs (Figure 6 – supplement figure 1C). Quantification of the total expression levels shows that the two mutants hCTFbaxA1254T and hCTFbax V1258A are expressed at higher levels relative to the reference construct hCTFbax mBS, suggesting that expression levels are sufficient for highlighting potential activity.

19. Figure 6B: Data must be collected in three independent experiments. it is not clear how this data was collected. Does "repeat" mean independent experiments?

Per the suggestion, the graph in Figure 6B now represents an n=3 independent experiments with statistical analysis.

20. Figure 7 A and B: Data must be collected in three independent experiments.

We have added data to obtain n=3-4 replicates for these experiments with statistical analysis.

21. Discussion could be trimmed as well.

The Discussion section has been streamlined as suggested. And a paragraph now comments the latest aGPCR cryoEM structures.

Reviewer 2:1. To address points 2, 5 and 6 in my public review, the authors should perform additional quantification of data and/or statistical comparisons.

We have addressed these points by adding quantification of expression, as permitted by the N-terminal or C-terminal epitope tag used in the constructs (Figure 3C; Figure 4C,D and E; Figure 4 supplement figure 3D; Figure 4 supplement figure 4C,D and E; Figure 5 supplement figure 1B; Figure 6 supplement figure 1C). For the concentration-response curves, we also added statistical analysis on the graphs, and all EC50 values are summarized in Figure 10.

2. To address points 1, 3, 4, and 7 in my public review, the authors should either add further data that supports their claims (if they want to maintain the interpretation of their data as currently written) or alternatively they should alter the discussion of their findings if some of their interpretations cannot be more strongly supported by additional data.

Please see the specific replies below in the Public Review section. We have edited the text as requested.

Reviewer 3:1. Title is confusing when referring to "… in extracellular loop 2 of the transmembrane region" and should preferably indicate "… in extracellular loop 2 and the transmembrane region".

We have edited the title per the suggestion.

2. The authors should analyze the role of the highly conserved Histidine residue in position 991 by generating at least one other substitution mutant with a conservative amino acid. A phenotypic rescue strategy could then be conducted by overexpressing the conservative mutant on the GPR116 knock-out background (previously generated by the research team). A parallel in vitro validation of the receptor mutant in HEK293 cells can be achieved to compare activation by GAP peptides, constitutive activity, expression and cleavage parameters all of which would help reinforce the authors' hypothesis on whether it is a lack of cleavage rather than the substitution of a highly conserved residue that is causing the described phenotypes.

We carefully considered the reviewers suggestion and offer the following two arguments in response to their inquiry:

3. Authors should provide a more detailed characterization of the GPR116 H991A/H991A mouse line. First, tissue expression of the mutated receptor should be assessed to discard the probability of a loss-of-function phenotype due to a lack of expression. Second, a direct comparison of AT2 cells secretion parameters should be included for the different genotypes (GPR116+/+, GPR116 -/-, GPR116 H991A/H991A), in order to gain direct mechanistic insights into genotype-linked phenotypes and evaluate the degree of loss-of-function phenotypes between GRP116-deficient mice and cleavage-deficient mice. Third, phospholipid homeostasis parameters should be tested further using AT2 cells from these mice in presence and absence of ATP stimuli (phospholipid secretion, phospholipid uptake).

First, we have generated additional data demonstrating comparable expression levels of H991A mRNA in primary AT2 cells to that seen in WT AT2 cells (Figure 2 —figure supplement 1D). Since we chose not to epitope tag the H991A knock-in construct in the mouse and lack a reliable antibody against GPR116, we are unable to quantitate and localize H991A protein in the animal model. However, we observed similar calcium responses in response to peptide stimulation in primary AT2 cells isolated from H991A/H991A mice compared to those seen in WT AT2 (Figure 2C and D). Taken together, these data strongly suggest that expression levels and membrane localization of H991A in the knock-in are similar to the WT protein in littermate control animals.

Second, we have generated additional data demonstrating that levels of the primary surfactant phospholipid, saturated dipalmitoylphosphatidylcholine (SatPC), are increased in the alveolar lavage fluid of H991A/H991A homozygous mice at levels comparable to those seen in aged matched GPR116-/- mice (Figure 2A). We believe that these data, together with: (1) similar histological findings in H991A/H991A and GPR116-/- mice; (2) similar mRNA levels in H991A and WT primary AT2 cells; and (3) similar peptide-induced calcium responses in H991A/H991A primary AT2 cells demonstrate that the H991A receptor is responsive in vitro but inactive in vivo.

Third, the primary goal of generating the cleavage-deficient mutant was to determine if receptor activation in vivo was dependent on cleavage at the GPS. Based on the data presented, we believe we have successfully demonstrated that the H991A mutant is expressed on the cell surface, is activated by exogenous peptide in a transiently transfected cell line and in primary AT2 cells, but is not functional in vivo based on the identical pulmonary phenotype of the H991A homozygous mouse line compared to the germline knock-out mouse line. We respectfully disagree that a complete characterization of the surfactant phospholipid secretion and uptake parameters in the H991A mouse line would add additional insight into the biology of the GPR116 beyond what is already presented in the manuscript.

4. Authors should determine expression levels for each receptor constructs in order to normalize their functional data accordingly. N-terminally Flag-tagged constructs provide an excellent opportunity to evaluate membrane exposure of mCTF receptors from the cell surface of HEK293 cells. This can be achieved by using flow cytometry (as performed in Supp Figure 7 for quantifying N-terminal mCherry signal) quantifying cell-surface fluorescence intensity of cell populations expressing respective receptor mutants which can be detected with anti-Flag antibody and fluorophore-coupled secondary antibody. Alternatively, a cell ELISA can be conducted using the same strategy but revealing with horseradish peroxidase substrates. As for C-terminally V5-tagged constructs, they can be analyzed also by flow cytometry coupled to immunofluorescence in order to obtain total protein expression levels as the positioning of the tag does not allow to make a distinction between subcellular compartments. For western blotting experiments, although band intensities can be visually approximated from western blot images, we highly recommend that authors make use of pixel intensity-based quantification software to monitor band intensities normalized using housekeeping proteins as loading controls.

As suggested, we have now included appropriate quantification for each set of constructs as follows:

We also edited the text as requested.

5. P.11, the authors conducted activation assays rather than binding assays and thus should refrain from inferring that " … set out to determine the region of the receptor to which the tethered agonist binds…". This sentence should be rephrased in order to better reflect the correlation that the authors try to establish using their structure-function assays strategy. Additionally, every mention of a binding site determination throughout the manuscript should be nuanced with this aspect in mind.

We rephrased several parts of the Results and Discussion sections accordingly including taking into account data presented in Figure 8D that suggests the possibility of peptide binding into the putative binding site.

6. In their systematic characterization of critical residues by conservative substitutions the authors have dismissed the T1240 position which leaves this seemingly systematic approach incomplete. In order to show consistency, we recommend that they generate a Threonine to Serine replacement at this position and include the analysis in the manuscript.

We have generated this construct as requested and demonstrate that the T1240S mutant is less responsive to peptide stimulation as compared to the parent mCTFbax construct. These data represented in Figure 4G of the revised manuscript.

7. In the generation of basal activity-null constructs ("bax" constructs), it is not clear why a different engineering strategy was adopted to generate the human versus mouse version. The authors should at least allude to it in the manuscript by shortly explaining or indicating if technical or circumstantial elements played a role in designing these different versions, or if other versions of hCTFbax where in fact designed but did not result in reduction in constitutive activity.

We added a comment in the Methods section with regard to the two different strategies. The hCTFbax construct was based on published studies on other aGPCRs, hypothesizing that deleting 6 amino acids was an effective approach to abolish the basal activity of the hCTF construct. The mCTFbax construct was generated in our previously published work (Brown et al., 2017) and we thought it to be appropriate to test hypotheses in this study.

8. For consistency purposes, the authors should evaluate all mGPR116 full-length versions of receptor mutants modifying key ECL residues with a version of full-length mGPR116 also containing an intracellular V5 epitope tag to discard any collateral effects due to the insertion of a tag that could potentially disrupt intracellular coupling events. This is particularly important giving that the human versions containing an intracellular V5 epitope (hFL versions) display expression differences (Supp Figure 6B).

Experiments in Figure 4 E,F and Figure 4 supplement figure 3A have been replicated, using the V5-tagged version of the full length receptor as a control. This permitted more accurate comparison of the expression levels between the reference construct and the mutants thereof. Quantification of total expression levels by flow cytometry were also added to complement the immunocytochemistry data.

9. In order to allow for a more comprehensive reading, the authors should briefly explain in the text what their strategy was when using different GAP peptides interchangeably for the different assays.

We commented in the Methods section the rationale for using GAP14 as reference peptide. GAP9 is the peptide with the minimum number of amino acids required for receptor activation comparable to that seen with GAP16 (previously published). GAP10 is more active than GAP9 and was used for specific studies looking at the importance of selective GAP amino acids.

Ex vivo and related experiments were performed originally with the GAP16 peptide. In additional experiments, GAP10 was used as a more active peptide.

10. The manuscript would greatly benefit from the inclusion of a table reporting the EC50 values obtained for all calcium transient experiments as this would allow for a more accurate comparison between the different conditions.

We have added Figure 10 reporting all EC50s and statistical analyses of the experiments.

11. Methodology: Immunocytochemistry- Please describe the reagents and their concentrations used for fixation, permeabilization and blocking methods. Calcium transients- Please briefly specify M3R activation protocol.

We have enhanced the Methods sections to include all three of these aspects.

12. P.15, Please correct the sentence as it contains a repetitive portion: “Conservative exchanges for the three other critical amino acids still yielded inactive receptors… all yielded inactive receptors”13. P16, please correct “… high probability of interactint…”.14. P.18, please correct "… have been have been…".15. P.20, please correct "… structural aspects aGPCRs activation, …".

Thank you for highlighting these errors – they have been corrected in the revision.

16. Figure 2 panels C and D, please include data obtained for control mice.

We have added additional data from wild-type control mice for the SatPC measurements (Figure 2A) and for calcium transient assays in primary AT2 cells (Figure 2C and D).

17. Figure 2: Please indicate data representation (mean), statistical error and analysis for panels C and D in figure legend

Added as requested.

18. For reproducibility purposes we recommend that the experimental set displayed in Figure 3 panel E for calcium and IP1 accumulation assays, be completed with at least n = 3 and after doing so, please indicate in figure legend the number of experimental repetitions, data/error type and statistical analysis conducted. Also, please indicate in the table the units used for IP1 accumulation assays as well as the respective concentration of indicated activating peptides that were used in this assay.

We have deleted the single IP1 data set and added calcium data to reach 3 independent experiments. Two peptides were however degraded in the meantime and could not be re-tested. We are presenting the data in a more concise way, including statistical analysis.

19. Figure 3 panel D, please indicate in figure legend the proper description of data representation (mean), error and statistical analysis as well as representativity (n of experiments). Also, please include in the panel the "dash line" which is mentioned in the figure legend.

We have edited the presentation of this figure and legend as suggested (Figure 3E)

20. Figure 3 panel B, please identify the mCTF lane as such.

Thank you for highlighting this error –mCTF was added to the figure as requested (Figure 3B).

21. Figure 8 panel A, please indicate which critical residues are represented by the other spheres at the top of TM3, middle and top of TM2, bottom of TM5, top of TM7 and/or TM2.

We have revised Figure 8 and improved the labelling of the new version.

22. Please provide scale bars for all microscopy images since some images appear to have been captured with a different zoom magnification (For example Supp. Figure 9).

We added the scale bars as suggested (Figure 4 supplement figure 1A, Figure 4 supplement figure 3AandC, Figure 5 supplement figure 1B, Figure 6 supplement figure 1B).

23. Figure legend for Supp Figure 6, please correct "HEK296" for "HEK293".

Thank you for highlighting this typo – it has been corrected in the revision (Figure 4 – supplement figure 3C).

24. Figure legend Supp. Figure 10, please indicate if results describe transient or stable HEK293 transfections.

This information was added for clarification (Figure 6 – supplement figure 1).

25. Supp. Figure 11, the table's second column title indicates "extracellular part of the NTF…" but should rather say "N-terminal part of the CTF…".

Thank you for this clarification. Text was edited as suggested (Figure 9).

26. Supp. Figure 11, please indicate with an asterisk the GPS cleavage-deficient aGPCR such as ADGRG5.

We added this information as suggested (Figure 9).

[Editors' note: further revisions were suggested prior to acceptance, as described below.]

We are most intrigued by the work you have reported about the importance of autoproteolysis for the GPR116 function in vivo. Please address the additional issues that the reviewers raised below in their comments including:– Please discuss the new GPR116 paper that describes a new ligand.

We added and discussed this reference in the context of our work in the Introduction section of our manuscript.

– Clarify the points about reproducibility.

We have added all the missing information and clarified the requested aspects. We also justified the single experiments for quantification of the receptor at protein level.

– Revise the title so the story can be better appreciated by the broad audience of eLife.

We have changed the title to: Regulation of pulmonary surfactant by the adhesion GPCR GPR116/ADGRF5 requires a tethered agonist-mediated activation mechanism.

– Rearrange the adhesion GPCR schematic in Figure 1 to correctly indicate the adhesion GPCR domains, including the GAIN domain and the GPS motif which is a peptide inside of the GAIN domain. Please also correct the same issue in the instruction.

We have modified the schematic in Figure 1 as requested and have edited the text referring to the schematic in the Introduction section.

– Provide a GPR116 TM/ peptide model based on the new adhesion GPCR structures. The current model is based on a class B family member.

We have generated a new model based on the cryo-EM structure of GPR110 and revised the results and Discussion sections accordingly.

Reviewer #2 (Recommendations for the authors):The authors were very responsive to the reviewers' comments and addressed all of the reviewers' concerns. The textual changes and new data added by the authors have significantly strengthened the manuscript.One additional comment I would like to make is that after the review of the original draft of this manuscript in Spring 2021, a paper was published reporting the secreted protein FNDC4 as a ligand for GPR116 (Georgiadi et al., Nat. Comm., 2021). This work is not mentioned in the present manuscript, and indeed the authors have a statement in the penultimate paragraph of their Introduction stating, "Still considered an orphan aGPCR, GPR116 has been proposed…"Is GPR116 still an orphan? It is understood that FNDC4 cannot be considered a definitive ligand for GPR116 until other groups have had a chance to confirm (or refute) the findings of Georgiadi and colleagues, but nonetheless, it seems that the work of Georgiadi et al. is highly relevant to the studies described in the present manuscript and therefore should at least be mentioned at some point.Along these same lines, it is terrific that the authors of the present manuscript made the effort to cite the recent wave of cryo-EM studies on aGPCRs and place their findings in the context of those new structural studies. In a similar vein, it would also be appropriate for the authors to cite the recently-published work of Georgiadi et al. and discuss how these findings may be relevant to the authors' own findings of the activation mechanism of GPR116.

Thank you for highlighting this point. We have included this FNDC4 reference in the introduction section of our manuscript. If GPR116 activation by FNCD4 is indeed confirmed by an independent group in the future, particularly in the context of pulmonary physiology, it could represent an interesting model to evaluate the dependence on the tethered peptide sequence for this agonist.

Reviewer #3 (Recommendations for the authors):I am satisfied with the authors' response to my original comments, but I have some suggestions that would need to be addressed for ease of reading, accuracy, and reproducibility purposes regarding the newly added data in this revised version:1. To avoid confusion, I would suggest adding "Cell surface expression (Mean fluorescence)" as the title for the Y axis of all graphs reporting results from surface exposed epitope in flow cytometry (Flag): Figure 3C, Figure 4 D, Figure 4 Supp figure 4D-E.

Thank you for the suggestion – we changed the titles accordingly as requested.

2. Likewise, I would suggest indicating "Total expression (mean fluorescence)" as the Y-axis title of all graphs reporting flow cytometry results from intracellular epitopes (V5/mCherry). Figure 4E, Figure 4 Supp Figure 4B.

Thank you for the suggestion – we changed the titles accordingly as requested.

3. Refrain from using the terms “membrane expression” when the methodology that has been used does not allow this determination exclusively since the immunocytochemistry protocol, as currently described in the methods section, involves a permeabilization step. This is the case in Figure 4 Supp figure 3A and 3C.

We reformulated these sentences as requested.

4. Please complete the newly added data with the corresponding information on reproducibility parameters (# of independent experiments, define error bars). Error bar descriptions and/or # of independent experiments are missing for various figures involving functional assays or for expression quantifications by western blot or flow cytometry (western blot quantifications do not display errors? Was it done just once? Justify if this is the case as stable cell lines can show decaying expression with time). These include: Figure 3F; Figure 3 Supp Figure 1 and 2; Figure 4; Figure 4 Supp Figure 1B,3B,3D, 4B-E; Figure 5A,B,E; Figure 5 Supp Figure 1B; Figure 6B-C; Figure 6 Supp Figure 1C; Figure 7; Figure 10 (error is SD or SEM?).

We added in all required figure legends the details on the presented data (average +/- SD; number of independent experiments and technical replicates). With respect to quantification of receptor expression by Western blot or flow cytometry: we performed these experiments in parallel to one of the functional experiments, to relate the two readouts. Two notable exceptions are Figure 4 Suppl Fig4D and Fig6 Suppl1C, which were run independently of functional assays; this flow cytometry experiment and western blot were therefore performed with freshly thawed vials of early passages, exactly for the reason mentioned by the reviewer.

5. It seems that flow cytometry experiments were performed only once, indicate if this is not the case.

Yes, these were single experiments. See above.

6. For flow cytometry experiments, indicate the number of events analyzed in the methods section.

We enhanced the methods section for flow cytometry with the number of cells measured for the various constructs.

7. The use of the term “biological replicates” seems incorrect throughout figure legends as they seem to indicate technical replicates instead. One experiment done at the same time is a n=1 that might include various replicates of each different groups, considered technical replicates; when the same experiment is done more than once at different times experiments are considered biological replicates. For example, the legend for Figure 4F indicates 3 independent experiments with 4 biological replicates but instead should read as 3 independent experiments done in quadruplicates or 4 technical replicates. Please revise where relevant and correct if this is the case or add if it is missing (For example Fig3F, Figure 3 Sup Figure 1, and others are missing replicates data).

We revised the figure legends accordingly.

8. Indicate where relevant when experiments are technical replicates instead of biological replicates. Add information about statistical significance where data represent biological replicates.

We revised the figure legends accordingly.

9. Because the data seems more relevant to the reader in this case, please interchange current panel E in Figure 4 with surface expression graphs for remaining mutants displayed in this figure depicted in Figure 4 Supp Figure 4 D-E.

We modified Figure 4 and related supplement Figure 4 as suggested.

10. Line 660, please correct the antibody incubation time indicating 230 min.

Done.

11. Indicate “% IP1 conversion” as the Y-axis title for all corresponding graphs.

Done (Figure 1D, Figure 3D, Figure3 Supplemental Figure 1A,B).

12. Note that no cell surface data expression is present for hFL-T1240A and is also missing for various hFL or hCTFbax or mCTF mutants. If data cannot be included at this stage, please include a brief comment in the manuscript to the effect of why this is the case in each section where the data is missing.

We have done our best to complement the Western blot data with immunocytochemistry to be transparent on the visible membrane localization of the receptor and associated mutants. We have now added a comment in each relevant section, when membrane quantification was not possible.

13. Figures 9 and 10 seem more suited for the supplementary section.

These data have been moved and are now referred to as Appendix-Table 1 and Appendix-Table 2.